

# IMPROVING A MULTI-GRAIN SIZE TOTAL SEDIMENT LOAD MODEL THROUGH A NEW STANDARDIZED REFERENCE SHEAR STRESS FOR INCIPIENT MOTION AND AN ADJUSTED SALTATION HEIGHT DESCRIPTION

Marine Le Minor[1,2], Dimitri Lague[2], Jamie Howarth[1], Philippe Davy[2]

[1]School of Geography, Environment and Earth Sciences, Victoria University of Wellington, Wellington, 6012, New Zealand
[2]Univ. Rennes, CNRS, Géosciences Rennes, UMR 6118, Rennes, 35000, France

*Correspondence to*: Marine Le Minor (marine.le-minor@univ-rennes.fr)

**Abstract.** Modelling sediment transport is important to understand how fluvial systems respond to climatic change or other
transient conditions such as catastrophic sediment release. In natural rivers, heterogeneity of sediment properties and
variability of flow regime result in different modes of transport that all contribute to the total sediment load. Le Minor et al.
(2022) presented a sediment transport law for rivers that extends from bed load to suspended load while being relevant for a
wide range of grain sizes but not specifically addressing the case of a distribution of grain sizes, which must also consider
the interactions between grain classes that are mainly important during the sediment erosion phase. If these interactions are
not properly considered, the model overestimates transport rates compared to measured ones. We present a new formalism
for the reference shear stress of multiple-size sediments, a parameter governing the onset of transport. We show that using a
reference shear stress standardized across datasets improves transport rate predictions made with the model of Le Minor et
al. (2022). We show that considering the bed roughness length as a reference transport height for single- and multiple-size
sediments significantly improves transport rate predictions. We also suggest that, for multiple-size sediments where the bed
surface is not fully mobile, the entrainment coefficient should include a dependency on the fraction of mobile grain sizes at
the bed surface, although data are insufficient to add this effect in a definite parameterization. Therefore, using a
standardized reference shear stress and a transport length adjusted with a common reference height across all sizes appear to
be two critical ingredients of a fully functional multi grain-size total sediment load model based on the disequilibrium length.
This adjusted model offers the potential to quantify grain-size specific sediment fluxes when different modes of transport
may be observed simultaneously, paving the way for more informed numerical modelling of fluvial morphodynamics and
sediment transfers.



## 1 Introduction

Transport over riverbeds made from sediment composed of heterogeneous grain sizes is complex due to concomitant bed load and suspended load transport. Capturing grain-size specific transport mode simultaneously is relevant for a variety of phenomena such as the total load partitioning under a wide range of water discharge (Turowski et al., 2010), sorting patterns at the bed surface, e.g., armoring and downstream fining (Paola et al., 1992; Powell, 1998; Viparelli et al., 2017), and gravel-sand transitions (Blom et al., 2017; Dingle and Venditti, 2023; Venditti and Church, 2014). These phenomena and being able

to model them accurately are important when investigating fluvial system response to climatic change or other transient conditions, such as the response to catastrophic sediment delivery from landsliding (Tunnicliffe et al., 2024).

To predict fractional transport rates of mixed-size sediments, i.e., for each grain size, it is important to account for grain size heterogeneity and grain size interactions, such as hiding-exposure effects that make the entrainment of fine fractions harder

(increased threshold of motion) and the entrainment of coarse fractions easier (lowered threshold of motion). Although sediment transport has been studied at grain-scale and reach-scale (e.g., Charru et al., 2004; Engelund and Hansen, 1967; Houssais and Lajeunesse, 2012; Lajeunesse et al., 2010; Meyer-Peter and Müller, 1948; van Rijn, 1984a, b), there has been, to our knowledge, only one attempt at a continuous transport law that extends from bed load to suspended load for a wide range of flow strengths and sediment grain size distributions introduced by Le Minor et al. (2022, model referred to as

LM2022 in this study).

LM2022 is a Multi Grain-Size Total Load Sediment Transport model that applies to various transport modes in both non-stationary and stationary regimes. It is based on the erosion-deposition formalism using a disequilibrium length, resulting in a spatial lag to reach transport saturation, which depends on flow condition and grain size (Charru, 2006; Daubert and

Lebreton, 1967; Davy and Lague, 2009; El kadi Abderrezzak and Paquier, 2009; Jain, 1992; Kooi and Beaumont, 1994; Lajeunesse et al., 2010; Lamb et al., 2008; Le Minor et al., 2022; Sklar and Dietrich, 2004). LM2022 succeeded in predicting the scaling between transport rate and excess shear stress for various transport modes, using two key elements: transport height and entrainment rate. Although the model succeeded in describing a continuum of transport rates from bed to suspended load for a given size, LM2022 had some limitations when the model predictions were compared to a variety of

experimental datasets for single and multiple-size bedload and total load transport. First, it tended to predict larger transport rates compared to measured ones for single and multiple-size sediment. The magnitude of this overestimation decreased for low shear stress but increased for high shear stress. This suggests that the transport length, entrainment rate, or both, may not be parameterized correctly since these two parameters set the magnitude and scaling of modelled transport rates with shear stress. Second, Le Minor et al. (2022) showed that the single-size erosion-deposition formulation based on the disequilibrium

length was not directly applicable to multiple-size sediments as the reference transport height was too dependent on individual grain sizes and not on the interactions between different grain sizes. They propose to use a common reference



transport height at the base of the transport layer. Preliminary results using the median grain size as this common reference transport height showed an improvement for multiple grain sizes.

Another explanation for the discrepancies in model predictions may be that input model parameters are not actually comparable between the experimental datasets, in particular when it comes to characterizing the incipient motion of sediment mixtures. Le Minor et al. (2022) use a critical shear stress formalism to characterize incipient motion in the entrainment rate. Critical shear stress means that for bed shear stress values lower than or equal to the critical value, there is no transport. Experimentally measuring the critical shear stress is very difficult due to the very low transport rates. An

alternative approach is commonly used instead, known as a reference shear stress (e.g., Shvidchenko et al., 2001; Wilcock and Crowe, 2003). The reference shear stress differs from the critical shear stress in that it is defined as the bed shear stress required to produce a given transport rate, and hence, for bed shear stress values lower than or equal to the reference value, there may be a very low non-zero transport rate. However, methods and estimates of the reference shear stress values vary between studies, resulting in a lack of consistency across the datasets used to evaluate the model in Le Minor et al. (2022).

Indeed, in Le Minor et al. (2022), reference shear stress values and empirical formulations derived from them were taken directly from the literature with no consideration of how they were measured (e.g., "by-eye" observations) or for which reference transport rate they were established. For instance, the reference transport rate considered by Wilcock and Crowe (2003) for mixed-size sediment was related to different entrainment probabilities from one grain size to another. This model contrasts with Shvidchenko et al. (2001), whose reference shear stress model relies on a unique entrainment probability.

Another issue, is that the reference shear stress measurement relates to a specific bed surface grain size composition and hydraulic conditions. The main limitations of existing definitions are that they were established based either on the grain size distribution of the initial bed surface, assuming that surface composition did not vary much between the initial and final run state (Shvidchenko et al., 2001) or the grain size distribution of the final bed surface averaged over several runs with similar initial bulk sediment (Wilcock and Crowe, 2003). These may not be comparable and could introduce inconsistencies in

model calibration or validation using different datasets, especially for bed load transport predictions.

In this study, we seek to improve on LM2022's model by: i) applying a standardized approach to determine the reference shear stress for incipient motion of single- and multiple-size sediments, ii) introducing a new formalism for the reference shear stress of individual fractions of multiple-size sediments based on the work by Shvidchenko et al. (2001) and Wilcock and Crowe (2003), and iii) present improvements of the entrainment rate and the transport length, two key model elements.

In doing so we seek to balance the objectives of accurately predicting single- and multiple-grain size bedload and total sediment load and developing a parsimonious model that is relatively easy to parameterize. In Sect. 2, we present the empirical equations of reference shear for incipient motion used in Le Minor et al. (2022), the key elements of LM2022's model, and improvements of its entrainment rate and transport length. In Sect. 3, we describe the datasets used for model

calibration and validation and the procedure to establish a new model of reference shear stress. In Sect. 4, we present results





for the new model of reference shear stress and transport rate predictions made considering the standardized threshold of motion and the modifications of the transport length. In Sect. 5, we discuss our findings and identify the key adjustments required to produce a Multi-Grain Size Total Sediment Load model that applies to single- and multiple-size sediments.

## 2 Model description and adjustment

### 2.1 Standardization of reference shear stress for incipient motion

In Le Minor et al. (2022), reference shear stress values used as model input were inconsistent across datasets since the methods used by the original authors were measured or calculated in different ways for single- and multi-grain size sediments (Table 1). We detail these differences below. As is usual, all equations are based on the dimensionless form of shear stress, i.e., the Shields stress $\theta_i$ [-]:

$$\theta_i = \frac{\tau}{\rho R_i g d_i} \quad (1)$$

where $R_i = \rho_{s,i}/\rho - 1$ [-] is the sediment specific gravity with $\rho_{s,i}$ and $\rho$ [kg m$^{-3}$] the sediment and water densities, respectively, $g$ [m s$^{-2}$] is the gravitational constant, $d_i$ [m] is the sediment diameter and $\tau$ [Pa] is the bed shear stress. Note that the subscript "c" is added when referring to the critical Shields stress $\theta_{c,i}$ and the subscript "r" is added when referring to the reference Shields stress $\theta_{r,i}$. We also refer to grain-size specific parameters using the "i"-subscript for the ith size.

For single-size sediments (Table 1), Le Minor et al. (2022) used critical Shields stress values published in the original studies or calculated using the modified form of the Shield's curve (Shields, 1936) by Soulsby and Whitehouse (1997):

$$\theta_{c,i} = \frac{0.3}{1 + 1.2 d_i^*} + 0.055 \left( 1 - \exp\left( -0.02 d_i^* \right) \right) \quad (2)$$

where $d_i^* = \sqrt[3]{\frac{R_i g d_i^3}{v^2}}$ [-] is the non-dimensional grain diameter ($v$ [Pa s] is the kinematic viscosity of water).



**Table 1: Summary of datasets and method employed to obtain values of Shields stress for reference shear stress in Le Minor et al.**
**(2022). In italics are two datasets that were added in this study (Shvidchenko et al., 2001; Shvidchenko and Pender, 2000a) .**

| | Dataset | Original method used in LM2022 to determine the reference Shields stress $\theta_{r,i}$ [-] | Method used to determine the bed roughness $z_0$ [m] |
|---|---|---|---|
| Single sizes | Meyer-Peter and Müller (1948, from the report by Smart and Jaeggi, 1983) | Empirical equation of Soulsby and Whitehouse (1997) | $3\,d_{90}/30$ |
| | Engelund and Hansen (1967, from the report by Guy et al., 1966) | Empirical equation of Soulsby and Whitehouse (1997) | Empirical equation of Nielsen (1992) to deal with bed forms |
| | Shvidchenko and Pender (2000a, from the doctoral thesis of Shvidchenko, 2000) | Empirical equations of Shvidchenko and Pender (2000a) | Empirical equation of Nielsen (1992) to deal with bed forms |
| | Lajeunesse et al. (2010) | Values provided in Table 3 of Lajeunesse et al. (2010) | $d_{90}/30$, value provided in Lajeunesse et al. (2010) |
| Multiple sizes | Shvidchenko et al. (2001, from the doctoral thesis of Shvidchenko, 2000) | Empirical equations of Shvidchenko et al. (2001) | Empirical equation of Nielsen (1992) to deal with bed forms |
| | Wilcock and Crowe (2003, from the experiment by Wilcock et al., 2001) | Empirical equations of Wilcock and Crowe (2003) | $3\,d_{90}/30$ |

For multiple-size sediments (Table 1), Le Minor et al. (2022) used critical shear stress values published in the original studies or reference Shields stress values calculated from empirical equations.

Note that we included two new datasets for our analysis compared to Le Minor et al. (2022): Shvidchenko and Pender (2000a) for single-size sediments and Shvidchenko et al. (2001) for multiple-size sediments. Both provide empirical equations of reference Shields stress that we used to predict transport rates with LM2022 as a reference case (Sect. 4).

In turn, two formalisms were used in this study for the multiple-size sediment datasets: Wilcock and Crowe (2003, model
referred to as WC2003 in this study), which is the most widely used when it comes to sediment mixtures, and Shvidchenko et al. (2001, model referred to as S2001 in this study). Each formalism consists of two equations, one for the reference Shields stress of the median size and one for the hiding function characterizing grain size interactions on a rough bed. They differ significantly in their approach to estimating the reference shear stress:

[1]. S2001's estimate is based on the incipient motion criterion (Shvidchenko et al., 2001; Shvidchenko and Pender, 2000a):



$$q_{s,i}^* = \frac{q_{s,i}}{F_i \sqrt{R_i\, g\, d_i^3}} \quad (3)$$

where $q_{s,i}^*$ [-] is the dimensionless transport rate per unit width or Einstein number (Einstein, 1950), $q_{s,i}$ [m³ s⁻¹] is the transport rate per unit width and $F_i$ [-] is the fraction on the bed surface. The incipient motion criterion relies on the 1:1 correlation between $q_{s,i}^*$ and the transport intensity $I_i$ [s⁻¹] that is the fraction of mobile grains at the bed surface per unit time.

[2]. WC2003's estimate is based on the reference transport criterion $W_i^*$ [-] (Parker et al., 1982b, a; Wilcock, 1988; Wilcock and Crowe, 2003):

$$W_i^* = \frac{R_i\, g\, q_{s,i}}{F_i\, u_*^3} \quad (4)$$

where $u_*$ [m s⁻¹] is the shear velocity.

 Empirical equations of the reference Shields stress have been established for sand-gravel mixtures using both the incipient motion criterion (Shvidchenko et al., 2001) and the reference transport criterion (Wilcock and Crowe, 2003) with reference values such as $q_{s,ref}^* = 10^{-4}$ (correlated to $I_{ref} = 10^{-4}$ s⁻¹) and $W_{ref}^* = 2.10^{-3}$, respectively.

The reference transport criterion of Parker et al. (1982a) was defined with no dependency on grain size for the purpose of a similarity collapse, meaning that transport rate data for individual fractions fall on the same curve and are not affected by grain size. This contrasts with the incipient motion criterion of Shvidchenko and Pender (2000a) that included a dependency on grain size since the transport intensity was defined as the ratio of the number of grain displacements to the number of grains available on the bed surface per unit time. Hence it may be interpreted as a measure of grain mobilization (ratio of mobilized grains to immobile grains) or a probability of entrainment (Shvidchenko et al., 2001; Shvidchenko and Pender, 2000a). Comparisons of these two criteria by Shvidchenko et al. (2001) reveal that the incipient motion criterion scaled by grain diameter results in similar mobilization rates across grain sizes contrary to the transport criterion. Consequently, the incipient motion criterion is better suited for developing a universal transport law applicable for single- and multiple-size sediments.

Recalling that the reference Shields stress is not defined in the same way in S2001 and WC2003, it is important to note that the authors both observe a dependency of $\theta_{r,i}$ with the heterogeneity of the grain size distribution at the surface of the bed. However, they measure this heterogeneity differently and obtain different empirical relationships that we describe below.






Shvidchenko et al. (2001) found a dependency of $\theta_{r,i}$ on median grain size $d_{50}$ [m], the grain size to median size ratio $d_i/d_{50}$ [-] and the mixture geometric log-standard deviation $\sigma_g$ [-]; i.e., $\sigma_g = \sqrt{d_{84}/d_{16}}$ (Nakagawa et al., 1982), where $d_{16}$ [m] and $d_{84}$ [m] are the grain diameters of the 16th and 84th percentile. For simplification, in this study, we refer to the mixture geometric log-standard deviation as the grain size sorting. In S2001, the grain-size specific reference Shields stress

writes:

$$\theta_{r,i} = \epsilon_i \frac{0.60}{a} s^{0.278} \quad (5)$$

where $a = -1.1\left(\log_{10}\left(1000\,d_{50}\right)\right)^3 + 4.8\left(\log_{10}\left(1000\,d_{50}\right)\right)^2 - 5.0\log_{10}\left(1000\,d_{50}\right) + 4.6$ [-] is the mobility factor, $\epsilon_i$ [-] is the hiding function and $s$ [-] is the channel-bed slope. Equation (5) was first established for the incipient motion of coarse single-size sediments by Shvidchenko and Pender (2000b). Parameter $a$ decreases from about 4.5 to 3 for grain sizes

between 1 and 5 mm and increases from about 3 to 5 for grain sizes between 5 and 100 mm.

The hiding function that corrects the reference Shields stress of the median size $\theta_{r,50}$ [-] from hiding-exposure effects writes:

$$\epsilon_i = \frac{\theta_{r,i}}{\theta_{r,50}} \begin{cases} \left(\dfrac{d_i}{d_{50}}\right)^{-e} & \text{if } \dfrac{d_i}{d_{50}} \leq 1 \\[2ex] \log_{10}\left(10\dfrac{d_i}{d_{50}}\right)^{-2.2} & \text{if } \dfrac{d_i}{d_{50}} \geq 1 \end{cases} \quad (6)$$

where $e = 2.0\,\sigma_g^{-0.10}\left(0.049\left(\log_{10}\left(1000\,d_{50}\right)\right)^3 - 0.26\left(\log_{10}\left(1000\,d_{50}\right)\right)^2 + 0.33\log_{10}\left(1000\,d_{50}\right) + 1.20\right) - 1.4$ [-]

is the hiding exponent. Parameter $e$ decreases for grain sizes between 1 and 5 mm and increases for grain sizes between 5 and 100 mm. The magnitude of parameter $e$ decreases with grain size sorting $\sigma_g$.

Wilcock and Crowe (2003) found a dependency of $\theta_{r,i}$ on the sand fraction $F_{sand}$ [-] and $d_i/d_{50}$. The grain-size specific reference Shields stress writes:

$$\theta_{r,i} = \left(\frac{d_i}{d_{50}}\right)^{-(1-b_i)} \theta_{r,50} \quad (7)$$

where $b_i$ [-] is the grain-size specific hiding exponent.

The reference Shields stress of median grain size writes:



$$\theta_{r,50} = 0.021 + 0.015 \exp\left(-20\, F_{sand}\right) \quad (8)$$

This equation means that the reference Shield stress starts at 0.036 for gravels and decreases down to 0.021 when the sand fraction is above 0.05.

The hiding exponent is as follows:

$$b_i = \frac{0.67}{1 + \exp\left(1.5 - \dfrac{d_i}{d_{50}}\right)} \quad (9)$$


We note that both models are rather complex, include a dependency on $d_i/d_{50}$ and express bed heterogeneity as either dependent on $d_{84}/d_{16}$ (S2001) that does not depend on a fixed grain size, or the sand fraction (WC2003) which is based on a specific grain size cutoff (2 mm). Moreover, S2001's reference Shields stress depends on bed slope to account for effects of relative depth (Shvidchenko and Pender, 2000a), but WC2003 does not. Relative depth is expected to alter turbulence,

velocity fields and thus transport intensity in the vicinity of incipient motion (Lamb et al., 2008; Prancevic and Lamb, 2015; Shvidchenko and Pender, 2000a). S2001 assumed that the reference shear stress increases with the channel-bed slope that is negatively correlated to the relative depth. Another difference pertains to the bed surface composition: S2001 was established using the bed surface composition at the initial state. In contrast, WC2003 considered the bed surface composition at the final state and averaged over runs with a similar initial sediment mixture. WC2003 was calibrated using

grain size distributions that are of about one order of magnitude wider than S2001: 0.1-64 mm (14 size classes) compared to 1-14 mm (8 size classes), respectively. The grain size sorting of the datasets used to establish the two formalisms overlap: 2.2-7.3 for WC2003 compared to 1.3-6.0 for S2001 (combination of datasets they used, 1.3 – 2.2 for their own data).

To resolve the inconsistency between reference shear stress models in Le Minor et al. (2022), we have reanalysed two of the

most complete multi-grain size sediment transport datasets (Shvidchenko et al., 2001; Wilcock and Crowe, 2003), as well as single-grain size datasets (Engelund and Hansen, 1967; Lajeunesse et al., 2010; Meyer-Peter and Müller, 1948; Shvidchenko and Pender, 2000a). In doing so our objectives were:

- to estimate reference shear stresses for each dataset using a standardized approach so they are comparable;
- to derive a new empirical model for the reference shear stress for multiple-size sediments, that includes a hiding-

exposure function with parameters describing the bed heterogeneity;
- to evaluate LM2022's model predictions using sediment transport datasets standardized by applying a common approach to calculate the reference shear stress.

Section 3 describes the detailed processing of the datasets that allow these three steps to be performed.



## 2.2 Key aspects of Le Minor et al. (2022)'s model

The erosion-deposition model is defined by two elements: the entrainment rate $\dot{e}_i$ [m s⁻¹] and the transport length $\xi_i$ [m] (Davy and Lague, 2009). This model has been extended to a spectrum of transport modes as well as to a spectrum of sediment grain sizes (Le Minor et al., 2022). Model elements relevant for this study are briefly described below and more details are given in Sect. S.1 of Supplementary Materials and in Le Minor et al. (2022).

According to Davy and Lague (2009), the transport length that links erosion and deposition may be parameterized with the thickness of the layer where most of the grains are transported, their transport velocity and their settling velocity. Le Minor et al. (2022) thus assume that the transport length writes:

$$\xi_i = \frac{h_{s,i}\,\overline{v_{s,i}}}{w_{s,i}} \quad (10)$$

where $h_{s,i}$ [m] is the sediment transport height, $\overline{v_{s,i}}$ [m s⁻¹] is the depth-averaged sediment transport velocity and $w_{s,i}$ [m s⁻¹] is the sediment settling velocity. The magnitude of the settling velocity is assumed to be equal to the terminal settling velocity calculated using the empirical equation of Ferguson and Church (2004) since it covers a wide spectrum of grain sizes.

The sediment transport length corresponds to the height reached by a grain after its ejection or detachment from the bed. Le Minor et al. (2022) write the transport height as:

$$h_{s,i} = \begin{cases} h_{salt,i} + \dfrac{h - h_{salt,i}}{r_{0,i}} \text{ if } T_i^* > 0 \\ 0 \text{ otherwise} \end{cases} \quad (11)$$

where $h_{salt,i}$ [m] is the saltation height, $r_{0,i}$ [-] is the gradient of vertical sediment distribution and $T_i^* = \tau/\tau_{r,i} - 1$ [-] is the transport stage with $\tau_{r,i}$ [Pa] the sediment reference shear stress. The saltation height writes:

$$h_{s,i} = \begin{cases} min\left(0.6\,d_i + 0.025\,d_i\,T_i^*,\, h\right) \text{ if } T_i^* > 0 \\ 0 \text{ otherwise} \end{cases} \quad (12)$$

To improve Le Minor et al. (2022) we propose a simplified formulation of the transport velocity assuming that sediment grains travel as fast as the water velocity averaged over the thickness of the layer where most transport occurs regardless of the transport mode, i.e., bed load or suspended load:



$$\overline{v_{s,i}} = \begin{cases} min\left( \dfrac{u_*}{\kappa}\left( \ln\left( \dfrac{h_{s,i}}{z_0} \right) - 1 + \dfrac{z_0}{h_{s,i}} \right), \overline{u} \right) & \text{if } T_i^* > 0 \\ 0 & \text{otherwise} \end{cases} \qquad (13)$$

where $\kappa$ [-] is the von Kármán constant, $\overline{u}$ [m s$^{-1}$] is the depth-averaged water velocity and $u_*$ [m s$^{-1}$] is the shear velocity. We observed that this simplified transport velocity formulation degrades neither the magnitude nor the scaling with the excess shear stress of single- and multiple-size transport rate predictions (see Fig. S.1 in Supplementary Materials). Maintaining the predictive power of the model with a simpler formulation is desirable because it makes the model easier to

parameterize. We thus use Eq. (13) to calculate the transport velocity from now on in this study.

Based on grain-scale dynamic studies (Charru et al., 2004; Lajeunesse et al., 2010), Le Minor et al. (2022) assume that the entrainment rate writes:

$$\dot{e}_i = F_i k_{e,i} \left( \tau - \tau_{c,i} \right) \qquad (14)$$

where $k_{e,i}$ [m$^2$ s kg$^{-1}$] is the entrainment coefficient. Here, we adjust LM2022 by replacing the critical shear stress by the reference shear stress:

$$\dot{e}_i = F_i k_{e,i} \left( \tau - \tau_{r,i} \right) \qquad (15)$$

For multiple-size sediments, Le Minor et al. (2022) was using Eq. (7)-(9) (Wilcock and Crowe, 2003) to calculate the critical

shear stress. In this study, we propose a new model for the reference shear stress for individual grain sizes in sediment mixtures and use that instead (see Sect. 4.1).

In the erosion-deposition model, the entrainment coefficient has the following form (Lajeunesse et al., 2010; Le Minor et al., 2022):

$$k_{e,i} = \frac{\alpha}{\rho_{s,i} w_{e,i}} \qquad (16)$$

where $\alpha$ [-] is an entrainment factor and $w_{e,i}$ [m s$^{-1}$] is the sediment ejection velocity. In Le Minor et al. (2022), the magnitude of the ejection velocity is assumed to be equal to the terminal settling velocity.

When there is no more spatial nor temporal variations of sediment load in the water column, the sediment load is at

equilibrium and writes:

$$q_{s,i}^{eq} = \xi_i e_i = F_i \frac{h_{s,i} \overline{v_{s,i}}}{w_{s,i}} \frac{\alpha}{\rho_{s,i} w_{s,i}} \left( \tau - \tau_{r,i} \right) \qquad (17)$$





In this formulation $\tau_{r,i}$ replaces $\tau_{c,i}$ which was used in the original LM2022 model.

The dimensionless form of this transport law (Einstein number) writes:

$$q_{s,i}^{eq*} = \frac{q_{s,i}}{\sqrt{R_i g d_i^3}} = F_i \alpha \frac{\rho}{\rho_{s,i}} \frac{h_{s,i}}{d_i} \frac{\overline{v_{s,i}} \sqrt{R_i g d_i}}{w_{s,i}^2} (\theta_i - \theta_{r,i}) \quad (18)$$

When the reference shear stress is not exceeded, we assume that the transport rate is so small that it can be neglected, and thus, the transport rate equals zero.

### 2.3 LM2022 model adjustment and calibration

As suggested by Le Minor et al. (2022), we modify the saltation height with an identical reference height for all sizes instead of a grain-size specific one. This is important as the higher the saltation height, the higher the grain velocity and, thus, the transport fluxes. Since the bed roughness $z_0$ plays a critical role in flow hydraulics and may be seen as the lower vertical limit of transport, we test the following modification of the saltation height:

$$h_{s,i} = \begin{cases} min(z_0 + 0.6 d_i + 0.025 d_i T_i^*, h) & \text{if } T_i^* > 0 \\ 0 & \text{otherwise} \end{cases} \quad (19)$$

In Le Minor et al. (2022), the measured density of moving sediment grains at the bed surface and the ratio of deposition to erosion time from Lajeunesse et al. (2010) and Houssais and Lajeunesse (2012) were used to obtain a value of the entrainment factor that was assumed to be constant. This allowed direct evaluation of model performance against experimental datasets without any prior calibration. However, this procedure is not applied in this study for two reasons: i) the density of moving sediment grains at the bed surface is not a parameter that is commonly measured in flume experiments and thus is lacking in most of the datasets used in this study, and, ii) the narrow range of hydraulic conditions and sediment properties explored by Lajeunesse et al. (2010) and Houssais and Lajeunesse (2012) does not provide us with a sufficiently board dataset for testing our hypotheses and highlighting phenomena potentially missing in the LM2022 model. Hence, as a first approach, we evaluate the effect of the standardized reference shear stress and new transport length using the original mobility factor of LM2022's model: $\alpha = \frac{\pi}{6} 37.64 = 19.71$. In a second step, we re-calibrate the model by estimating an empirical value of $\alpha$ in response to model adjustments. We calculate it as the median value of the population of values of $\alpha$ for each measured transport rates as follows:





$$\alpha = \frac{q^{eq}_{s,i,measured}}{F_i \, \xi_{s,i,predicted} \, \dfrac{1}{\rho_{s,i} \, w_{e,i}} \left( \tau - \tau_{r,i} \right)} \quad (20)$$

**3 Data and methods**

**3.1 Datasets**

Six datasets from flume experiments on single- and multiple-size sediments were used in this study (Table 1) for: (i) model comparison of the original LM2022's model, (ii) calibration of the new reference shear stress, and (iii) calibration of the new version of LM2022's model, specifically the new reference shear stress for incipient motion and the entrainment factor $\alpha$. See Sect. S.2 of Supplementary Materials for details on the range of values tested for each dataset.

Contrary to Le Minor et al. (2022), the dataset of Houssais and Lajeunesse (2012) was not considered in this study since values of depth-averaged water velocity used to calculate the transport length are lacking. For the dataset of Meyer-Peter and Müller (1948), only the data where the median size $d_{50}$ [m] is the same as the 90th-percentile size $d_{90}$ [m] were used, i.e., with a single grain size (5.21 mm and 28.65 mm).

In Le Minor et al. (2022), the sensitivity analysis revealed that the transport rate predictions were affected by the bed roughness $z_0$ fed to the model and hence that the impact of bed forms on the bed roughness should be accounted for. Details on the bed roughness considered for each dataset are given in Table 1.

**3.2 Datasets processing to estimate the reference shear stress**

As in Le Minor et al. (2022), for all the datasets, the same method as the one mentioned in Wilcock and Crowe (2003) was applied to calculate values of bed shear stress corrected for sidewall effects (Chiew and Parker, 1994; Vanoni and Brooks, 1957). In the following work, we hypothesize that the reference Shields stress corresponds to the bed shear stress that produces a reference transport intensity of $10^{-4}$ s$^{-1}$ and, thus, Einstein number of $10^{-4}$ (Shvidchenko and Pender, 2000a). This value corresponds to only rare movements of grains at the bed surface (Kramer, 1935) and transport is considered negligible below it. Reference Shields stress values were obtained by plotting the transport intensity data as a function of grain-size specific Shields stress. We then use an empirical fit such as $\log\left(q^*_{s,i}\right) = a + \dfrac{b}{\theta_i}$ where $a$ and $b$ are two constants (Fig. 1) instead of a power law $\log\left(q^*_{s,i}\right) = a + b \log\left(\theta_i\right)$. The power law is suitable for constant scaling of the dimensionless





transport rate (Einstein number) with the Shields stress. However, this scaling is not constant for all the datasets we use, especially for the finer grain sizes, where it varies from about 1.5 for bed load transport to 2.5 for suspended load. Applying an exponential fit better captures this varying scaling. With the above equation, the reference Shields stress is:

$$\theta_{r,i} = \frac{b}{\log\left(q_{s,ref}^{*}\right) - a} \quad (21)$$

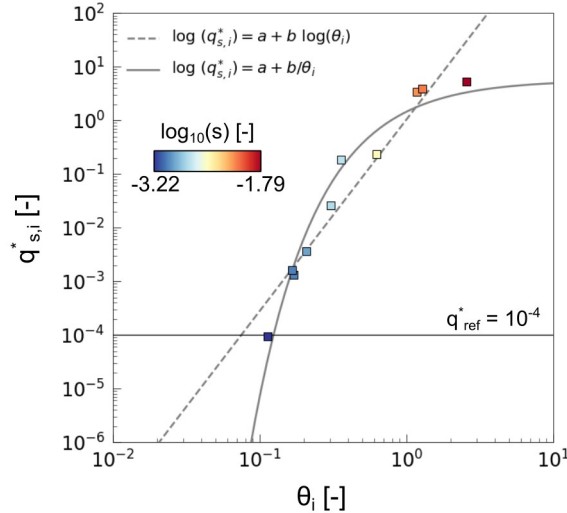

**Figure 1: Example of fit to data points using two interpolation methods. Dimensionless transport rates, i.e., values of Einstein parameter, for data of Wilcock and Crowe (2003) (finest grain size 210-500 μm and highest initial sand fraction 34%), are plotted as a function of Shields stress. Correlation coefficient of 0.92 for the linear fit and of 0.96 for the more complex fit used in this study.**

The automatic fitting procedure requires at least three data points and that the resulting fit has a trend similar to the one expected (increasing trend in the Einstein number with increasing Shields stress). Otherwise, the reference shear stress is considered as unknown and the corresponding measurements unexploitable. Ultimately, we obtained 40 estimates of the reference shear stress out of 52 for single-size sediments and 182 out of 240 for multiple-size sediments.

For the single-size data, one value of reference shear stress was calculated per grain size. For the multiple-size data, one value of reference shear stress was calculated per grain size and per mixture (initial state). For the single-size data of Shvidchenko and Pender (2000a) and the multiple-size data of Shvidchenko et al. (2001), several runs were carried out for a constant channel-bed slope as well and hence we calculated one value of reference shear stress per channel-bed slope, to evaluate its influence.



### 3.3 Procedure to establish a new formalism of reference shear stress for multiple-size sediments

For the multiple-size sediments, we went one step beyond the measurements of reference shear stress by building a new model of incipient motion of individual sizes that combines the formalisms of Shvidchenko et al. (2001) and Wilcock and
Crowe (2003).

A preparation of the data was needed to extract the necessary parameters from the two datasets:
 1. Each dataset was subdivided according to the grain size, initial sediment mixture and channel-bed slope when applicable (single-size data of Shvidchenko and Pender, 2000a, and multiple-size data of Shvidchenko et al., 2001).
2. Final surface properties ($d_{50}$, $\sigma_g$ and $F_{sand}$) were averaged over the runs carried out for a given initial sediment mixture and channel-bed slope when applicable.
 3. For each initial sediment mixture and channel-bed slope when applicable, we had a series of grain sizes with their calculated reference Shields stress and corresponding final surface properties. Linear interpolations were conducted over this series of grains sizes in log-scale to determine the reference Shields stress $\theta_{r,50}$ of the median size $d_{50}$ estimated in step 2.

Once we have the standardized datasets, we proceed in developing an empirical model for $\theta_{r,50}$ as function of bed heterogeneity, exploring the potential influence of $d_{50}$, $\sigma_g$ and $F_{sand}$, and we then develop a new hiding-exposure model.

### 4 Results

### 4.1 Standardized reference shear stress for incipient motion

Figure 2a shows that $\theta_{r,50}$ decreases significantly with final surface grain size sorting ranging from ~0.03 for very large grain size heterogeneity and converging to values ranging between 0.05-0.06 for nearly uniform grain size mixtures ($\sigma_g \approx 1$). This is consistent with the expected trend that was reported for non-uniform sediments using the transport rate criterion by Patel and Ranga Raju (1999) and Patel et al. (2010) but the magnitude is slightly larger.

We found that the reference Shields stress of the median grain size in the mixture can be expressed as (Fig. 2a):

$$\theta_{r,50} = 0.060\,\sigma_g^{-0.469} \quad (R^2 = 0.69) \quad (22)$$



Figure 2a shows that there is no clear influence of $d_{50}$ on $\theta_{r,50}$ and that Eq. (22) holds for S2001 and WC2003. Using a

functional relationship similar to WC2003, we found that the correlation between $\theta_{r,50}$ and the sand fraction was weaker (

$\theta_{r,50}=0.049\exp\left(-0.97\,F_{sand}\right)$, $R^2=0.30$), so that we do not use $F_{sand}$ subsequently.

We seek to develop a simple hiding function following previous studies (e.g., Parker and Klingeman, 1982; Wilcock and Crowe, 2003):

$\theta_{r,i}=\left(\dfrac{d_i}{d_{50}}\right)^{-\gamma_i}\theta_{r,50}$ (23),

in which $\gamma_i$ could depend on $d_i/d_{50}$ as in WC2003, or on grain size sorting $\sigma_g$ as in S2001. For this we compute individual values of $\gamma_i$ calculated as $\gamma_i=-\log\left(\theta_{r,i}/\theta_{r,50}\right)/\log\left(d_i/d_{50}\right)$, for all experimental data. Figure 2b shows the variation of $\gamma_i$ as a function of $d_i/d_{50}$ and $\sigma_g$. The range of $\gamma_i$ obtained varies between 0.1 to 1.2. Three negative points were ignored. Recall that $\gamma_i=1$ means perfect equal mobility, as the reference shear stress is identical for all size classes. If $\gamma_i<1$, the

reference shear stress increases with grain size resulting in classical size selective entrainment, while $\gamma_i>1$ predicts that the reference shear stress decreases with grain size. Figure 2b shows that $\gamma_i$ decreases significantly with the grain size sorting leading to a pronounced size selective entrainment, while as the final surface tends to a uniform grain size ($\sigma_g\approx 1$), equal mobility conditions dominate. We fit a power law to account for this dependency (Fig. S2a in Supplementary Materials). The residuals show a dependency with $d_i/d_{50}$ (Fig. S2b in Supplementary Materials) that we adjust with a function similar to

Wilcock and Crowe (2003) (Eq. (9)). This leads to a new formulation of $\gamma_i$:

$\gamma_i=1.275\,\sigma_g^{-0.789}\left(1.461-\dfrac{0.859}{1+\exp\left(1-\dfrac{d_i}{d_{50}}\right)}\right)$ ( $R^2=0.74$ ) (24)

We propose a hiding function formulation that combines both versions of the Shvidchenko et al. (2001) and Wilcock and Crowe (2003) empirical models and parameters: the variations of the exponent of hiding-exposure with both the grain size

sorting and the grain size to median size ratio are shown in Fig. 2b.



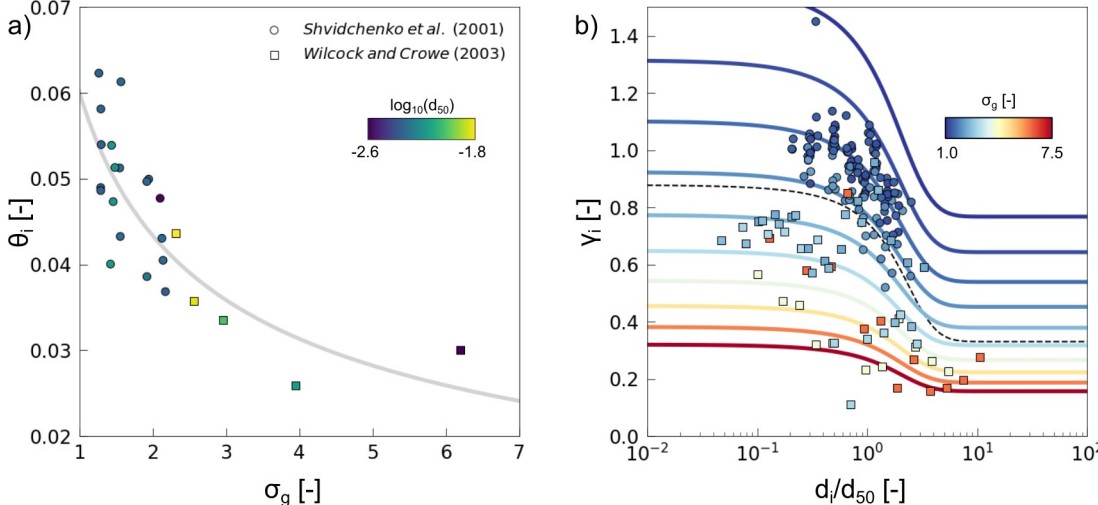

**Figure 2: New formalism of reference shear stress for incipient motion of multiple-size sediments. a) Variations of reference**
**Shields stress of the median size as a function of the final surface grain size sorting. The grey line corresponds to Eq. (22) (**
$R^2 = 0.69$**). Markers are colored according to the median grain size. b) Variations of exponent in hiding function as a function of**
**grain size to median size ratio. Markers are colored according to the final surface grain size sorting. Equation (24) ($R^2 = 0.74$) is**
**plotted for ten values of final surface grain size sorting. The dashed line corresponds to the equation of Wilcock and Crowe (2003).**

The former two formalisms of Shvidchenko et al. (2001) and Wilcock and Crowe (2003) cannot be shown on the same plot
as our new formalism since they are not equivalent, i.e., different variables and methods were used to parameterize and
evaluate the reference shear stress. Figure 3 compares the modelled reference shear stress values calculated using Eq. (22)
and (24) and the interpolated ones obtained using the approach shown in Fig. 1. Our approach manages to predict both the
values we have interpolated with the method presented in section 3.2 for Shvidchenko et al. (2001) and Wilcock and Crowe
(2003) data better ($R^2 = 0.92$) than the equations of Shvidchenko et al. (2001, $R^2 = 0.69$) and Wilcock and Crowe (2003,
$R^2 = 0.64$).





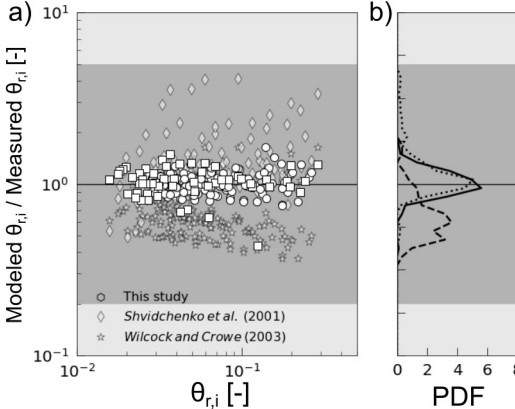

**Figure 3: Comparison between our new formalism and the ones of Shvidchenko et al. (2001) and Wilcock and Crowe (2003). a)**
**Comparison of predicted and measured values of reference Shields stress. b) PDF of the residuals. The solid, dashed and dotted**
**lines correspond to the PDF of residuals obtained with the set of equations from this study, Shvidchenko et al. (2001) and Wilcock**
**and Crowe (2003), respectively.**

## 4.2 Impact of model changes on transport rate predictions

### 4.2.1 Original model parameterization of Le Minor et al. (2022)

We use the Le Minor et al. (2022) model results as a reference case against which we compare new model adjustments (Fig.
4a to 4d). The original value for the entrainment factor in LM2022's mode was $\alpha = 19.71$. This model parameterization
produced transport rate predictions that had residuals (ratio of transport rate predictions to flume observations) that exhibit a
decreasing trend with the excess of Shields stress between $10^{-3}$ and $10^{-1}$, and a slightly increasing trend below $10^{-3}$ and above
$10^{-1}$. 53% and 65% of the single-size experimental data were predicted within a factor of 5 and 10, respectively. 36% and
51% of the multiple-size experimental data were predicted by our model within a factor of 5 and 10, respectively.

### 4.2.2 Standardized reference shear stress for incipient motion

We predict transport rates with interpolated values of reference shear stress for single-size sediments (procedure described in
Section 3.2) and calculated values of reference shear stress for multiple-size sediment due to changes in bed composition
(Eq. (22) and (24)).

First, we do not adjust $\alpha$ (Fig. 4e to 4h). Compared to Le Minor et al. (2022), for the single-size sediments, the decreasing
trend with the excess of Shields stress between $10^{-3}$ and $10^{-1}$ is considerably attenuated, while the increasing trend below $10^{-3}$
and above $10^{-1}$ remains (Fig. 4e). For the multiple-size sediments, the decreasing trend is slightly attenuated (Fig. 4g). The





scattering of residuals is reduced for the single-size sediments, while it does not change for multiple-size sediments (Fig. 4e-h). In addition, 59% and 80% of the single-size experimental data are predicted by our model within a factor of 5 and 10, respectively. 39% and 55% of the multiple-size experimental data are predicted by our model within a factor of 5 and 10, respectively. Note that the residuals are not centred on one when considering the original entrainment factor of LM2022's

model, and thus, a re-calibration of this factor is required to improve the magnitude of predicted transport rates.

Re-calibrating the entrainment factor for single- and multiple-size sediments using the approach described in Section 2.3 gives an entrainment factor $\alpha = 4.61$. 81% and 89% of the single-size experimental data (~20% improvement compared to Le Minor et al. (2022) are predicted within a factor of 5 and 10, respectively. 54% and 73% of the multiple-size experimental

data (~15% improvement) are predicted within a factor of 5 and 10, respectively. Hence, the standardization of the threshold of motion enhances both single and multiple-size sediment transport predictions once the entrainment coefficient is readjusted.

### 4.2.3 Standardized reference shear stress and common reference transport height

Here we first keep the original parameterization of Le Minor et al. (2022) ($\alpha = 19.71$ but using the new reference shear stress and setting the bed roughness as the minimum saltation height across all grain sizes (Eq. (19)). Figure 4i shows that for the single-size sediments, the decreasing trend with the excess of dimensionless shear stress between $10^{-3}$ and $10^{-1}$ remains, as does the increasing trends below $10^{-3}$ and above $10^{-1}$ (Fig. 4i). For the multiple-size sediments, the decreasing trend is attenuated (Fig. 4k). Scattering of residuals is reduced for both single- and multiple-size sediments (Fig. 4i-l). In addition,

56% and 76% of the single-size experimental data are predicted by our model within a factor of 5 and 10, respectively. This is a marginal improvement. 32% and 48% of the multiple-size experimental data are predicted by our model within a factor of 5 and 10, respectively, which is worse than the original model. The residuals are not centred on one when considering the original entrainment factor of LM2022'model, similar to the outcome of the step above.

Re-calibrating the entrainment factor for single- and multiple-size sediments at the same time as described in Section 2.3 gives an entrainment factor $\alpha = 2.51 \pm 28.92$ and results in the best predictions of the transport rate magnitudes (Fig. 5): 79% and 92% of the single-size experimental data (~25% improvement compared to Le Minor et al. (2022) are predicted by our model within a factor of 5 and 10, respectively. 62% and 81% of the multiple-size experimental data (~20% improvement compared to Le Minor et al. (2022) are predicted by our model within a factor of 5 and 10, respectively. The

combination of the standardized reference shear stress, common reference height and re-calibrated entrainment factor thus gives substantial improvements for both single and multiple-size sediment transport compared to the original LM2022 model version.



*Le Minor et al. (2022)* **- Reference case**

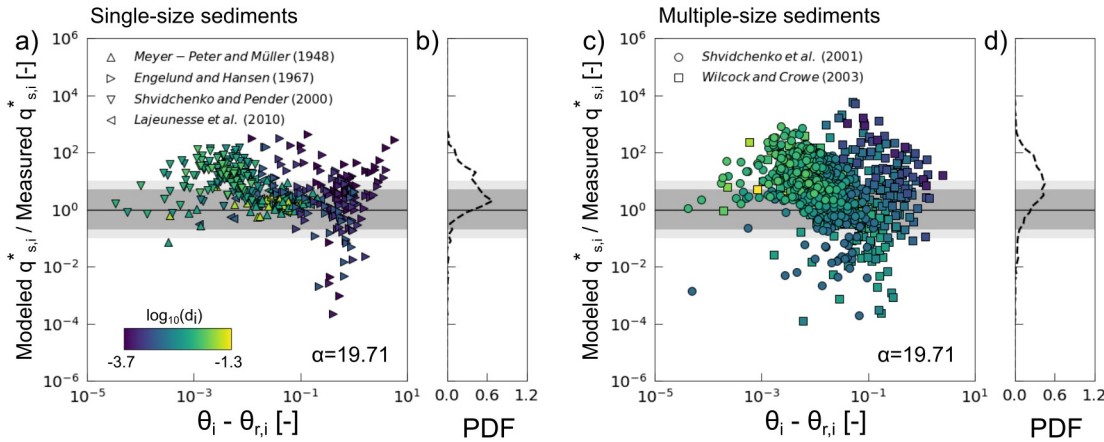

*Le Minor et al. (2022)* **+ Standardized reference shear stress**

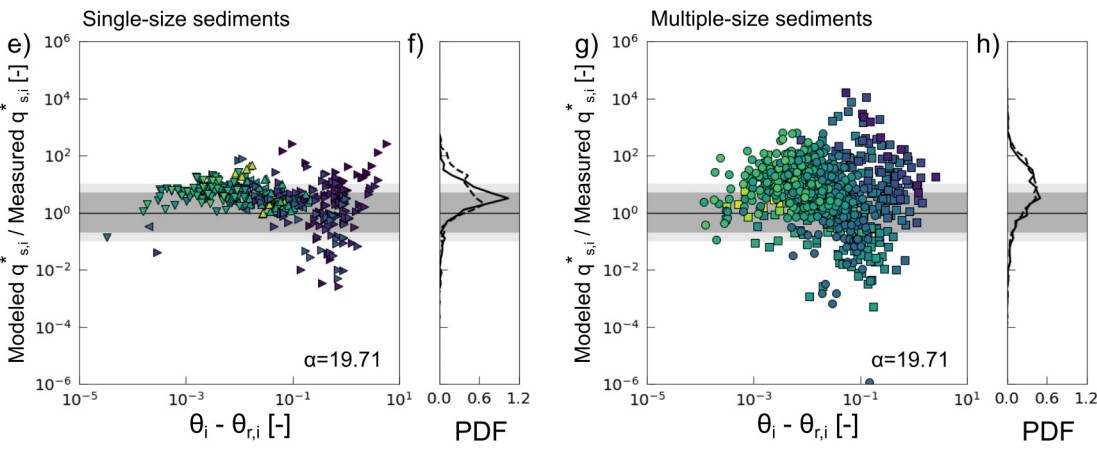

*Le Minor et al. (2022)* **with Standardized reference shear stress + Adjustment I**

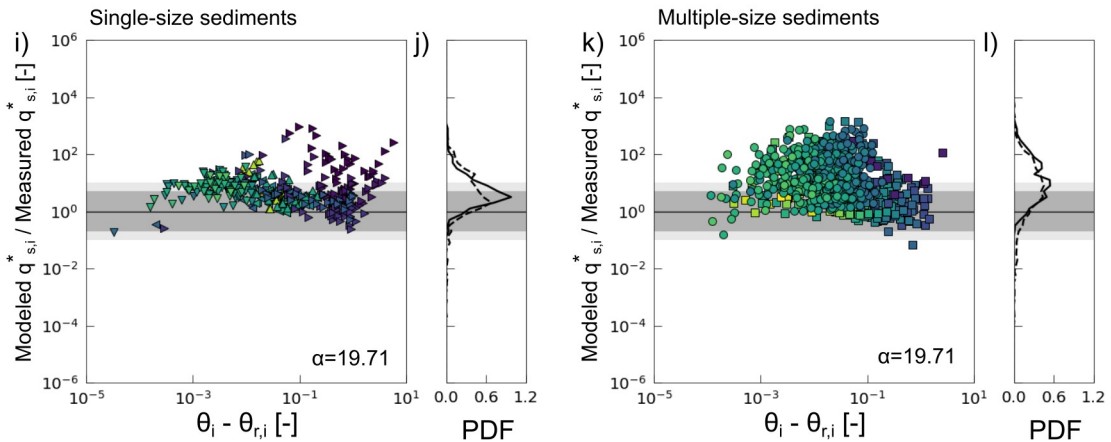




**Figure 4: Effects of standardized incipient motion and adjusted transport height on predictions of total load transport rates at equilibrium using the original entrainment factor of LM2022's model. a-d) Predictions with the same critical shear stress values as of Le Minor et al. (2022) and the original parameterization of LM2022's model. e-h) Predictions with the original parameterization of LM2022's model and the standardized reference shear stress values. i-l) Predictions with the LM2022's model**

**adjusted with the bed roughness length as reference height across all grain sizes and the standardized reference shear stress values. The ratio of model predictions to flume observations is plotted against the dimensionless excess of shear stress (Shields stress) along with the probability density function (PDF) of the residuals for single-size sediments (a-b, e-f, i-j) and multiple-size sediments (c-d, g-h, k-l). For the PDF, the dashed and solid lines correspond to predictions with the original LM2022's model and the adjusted model, respectively. The dark and light gray areas correspond to measured values that are predicted within a factor**

**of 5 and 10, respectively.**

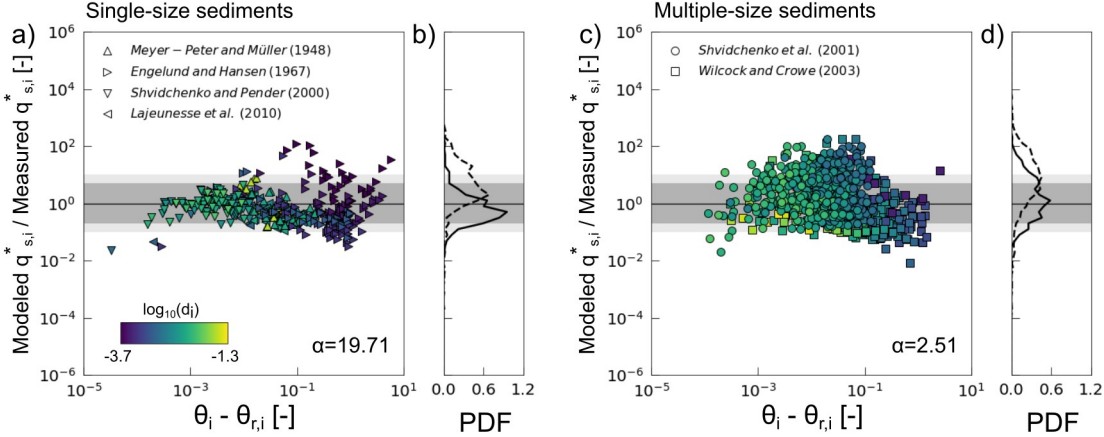

**Figure 5: Effects of standardized incipient motion and adjusted transport height on predictions of total load transport rates at equilibrium using the re-calibrated entrainment factor. Predictions with the LM2022's model adjusted with the bed roughness**

**length as reference height across all grain sizes and the standardized reference shear stress values for single- (a-b) and multiple-size sediments (c-d). The ratio of model predictions to flume observations is plotted against the dimensionless excess of shear stress (Shields stress) along with the probability density function (PDF) of the residuals for single-size sediments (a-b) and multiple-size sediments (c-d). For the PDF, the dashed and solid lines corresponds to predictions with the original LM2022's model and the adjusted model, respectively. The dark and light gray areas correspond to measured values that are predicted within a factor of 5**

**and 10, respectively.**



## 5 Discussion

### 5.1 Reference shear stress for incipient motion

To improve on the inconsistency of reference shear stress estimates in Le Minor et al. (2022), we present a reanalysis of two of the largest and well documented surface based multiple-size transport datasets: Shvidchenko et al. (2001) and Wilcock and Crowe (2003).

We show that $\theta_{r,50}$ decreases as a power-law with final surface grain size sorting (Eq. (22)). This dependency was not considered by Shvidchenko et al. (2001) or Wilcock and Crowe (2003). Shvidchenko et al. (2001) expressed the reference Shields stress of the median size as the one for a uniform coarse sediment that varies with grain size and channel-bed slope (Shvidchenko and Pender, 2000a). As for Wilcock and Crowe (2003), the Shields stress of the median size only depends on the sand fraction on the bed surface. The dependency on channel bed slope introduced by Shvidchenko and Pender (2000a) is a proxy for the relative depth. We explored the potential effect of relative depth on $\theta_{r,50}$, and found that it was of lower magnitude than the final surface grain size sorting ($\theta_{r,50}=0.046\,\sigma_g^{-0.471}\left(h/d_{50}\right)^{0.089}$, $R^2=0.72$, $p=0.131$, and $\theta_{r,50}=0.108\,\sigma_g^{-0.446}\,s^{0.122}$, $R^2=0.74$, $p=0.057$). Combining final surface grain size sorting and relative depth with the same hiding function (Eq. (24)) does not improve significantly the quality of the predictions ($R^2=0.92$ when comparing modelled and interpolated values of reference shear stress). More data are needed to distinguish the role played by each parameter.

The hiding-exposure function (Eq. (24)) is a key component of multiple-size sediment transport since it quantifies how interactions between grain sizes such as sheltering of fine grains and exposure of coarse grains increase and lower the reference shear stress, respectively. Our empirical equation for the hiding exponent shows a dependency on both the grain size sorting and grain size to median size ratio (Eq. (24)). Our new formalism is similar to Shvidchenko et al. (2001) since it relies on the incipient motion criterion but differs from it because it is based on the final bed composition instead of the initial one. Our new formalism is similar to Wilcock and Crowe (2003) since it is based on the final bed surface composition but differs from it because it relies on the reference transport criterion. Final bed surface properties were averaged over runs with similar initial bulk mixture and channel-bed slope. This averaging procedure smooths the surface properties at the final state of experimental runs characterized by variable hydraulic conditions with the same initial sediment mixture. The smoothing is more important for large initial grain size sorting (increasing values of $\sigma_g$) since the range of surface properties at the final state is wider. Although our new model for reference shear stress does not account for this (intra-subset) heterogeneity, the averaging procedure seems sufficient to link bed surface composition at the final state and sediment transport at equilibrium as shown by the quality of the residuals (Fig. 3). In addition, we propose a formalism of intermediate



complexity between Wilcock and Crowe (2003) and Shvidchenko et al. (2001), while covering a broader range of than each study taken individually. Our model is slightly more complex than Wilcock and Crowe (2003) but has lower residuals, and explicitly includes grain sorting as an important controlling factor (Fig. 2). At the same time, our model is much simpler than

Shvidchenko et al. (2001) but also has lower residuals. On balance, we argue it is the best model because it provides intermediate complexity with better predictive power over a wider parameter space.

Note that the new formalism presented in this study has been established for sand-gravel mixtures with grain sizes ranging from 0.1 to 64 mm. Equation (22) for the reference shear stress of the median size applies to mixtures with gravel as median

size (surface final median size between 2.5 and 16.4 mm).

## 5.2 Bridging the gap between the reference shear stress of single- and multiple-grain size transport

Based on our new formalism (Eq. (22)), the reference Shields stress of median size tends towards 0.06 for nearly uniform grain size mixtures ($\sigma_g \approx 1$). The reference Shields stress values determined with the method presented in Fig. 1 for single-

size datasets used in this paper range from about 0.01 to 0.1. A reference Shields stress of 0.06 is consistent with values obtained for gravel for the dataset of Shvidchenko and Pender (2000b, a). Thus, our new formalism bridges the gap between reference shear stress of single-size gravel and multiple-size sediments. However, it overestimates by up to a factor six the values reported for medium and coarse sand for the dataset of Lajeunesse et al. (2010) and underestimates by a factor two the values reported for fine sand for the dataset of Engelund and Hansen (1967). Expressing the reference Shields stress of

median size as a function of sand fraction as in Wilcock and Crowe (2003) such as $\theta_{r,50}=0.049\exp\left(-0.97\,F_{sand}\right)$ ( $R^2=0.30$) could improve its suitability for single-size sediments since it tends towards 0.018 for sand mixtures. However, this formulation was not as statistically significant as the grain size sorting so further investigations are required to explore this option.

Some studies have pointed out the potential role of the slope dependency on incipient motion (Lamb et al., 2008; Recking, 2008; Shvidchenko and Pender, 2000a), and found that the reference shear stress increases with the channel-bed slope due to a concomitant decrease in the water depth to grain size ratio, i.e., relative depth. This trend stems from grains that occupy a large portion of the water column and, in turn, have a larger resistance to the flow.

More sediment mixtures should be tested to find out how the reference shear stress of the median size varies when the median size becomes finer (finer than gravel), the grain size sorting is higher and for a wider range of channel-bed slope. The dependence of the hiding function on the relative depth and the properties of the grain size distribution should also be investigated.



**5.3 LM2022 model adjustment**

Our results show that standardizing the reference shear stress for incipient motion has a positive impact on transport rate predictions, especially after recalibration of the entrainment factor. Our study supports the fact that the original ingredients of LM2022's model hold since it is able to predict the magnitude and scaling of transport rates when the reference shear stress is properly standardized.

As suggested in Le Minor et al. (2022), our results show that a key adjustment of LM2022's model is the addition of the bed roughness length as the minimum transport height (Eq. (19)). We assume that the bed roughness length is a hydraulic boundary that has meaning for single and multiple-size sediment transport, i.e., negligible transport below, and all the entrained grains leave the bed from this height whatever their size. This is important as it impacts the calculation within LM2022 of the mean flow velocity experienced by a grain.

Here we explore how LM2022's improvements impact the predictions for suspended and total load, using the dataset by Engelund and Hansen (1967). The comparison between the performance of LM2022's model and our adjusted model shows that the modifications presented in this study improve total load predictions as well (Fig. 6). As discussed in Le Minor et al. (2022), bed forms likely affect the bed roughness that is a model input for the transport height and transport velocity and consequently may impact the transport rate predictions. To account for bed forms, as in LM2022's model, we used the equation of Nielsen (1992) to calculate the bed roughness for the experiments of Engelund and Hansen (1967), Shvidchenko and Pender (2000a) and Shvidchenko et al. (2001) where bed forms development was identified on the bed. Only averaged bed form dimensions have been reported for the first two studies, while for the latter, they were lacking for some runs. Considering no bed forms in the case of Engelund and Hansen (1967), i.e., a bed roughness of $3d_{90}/30$, reduces residual scattering and improves transport rate predictions for total sediment load (Fig. 6). It is likely that a better knowledge of the bed roughness would increase model performance, although this parameter is difficult to measure due to its spatial-temporal variations.





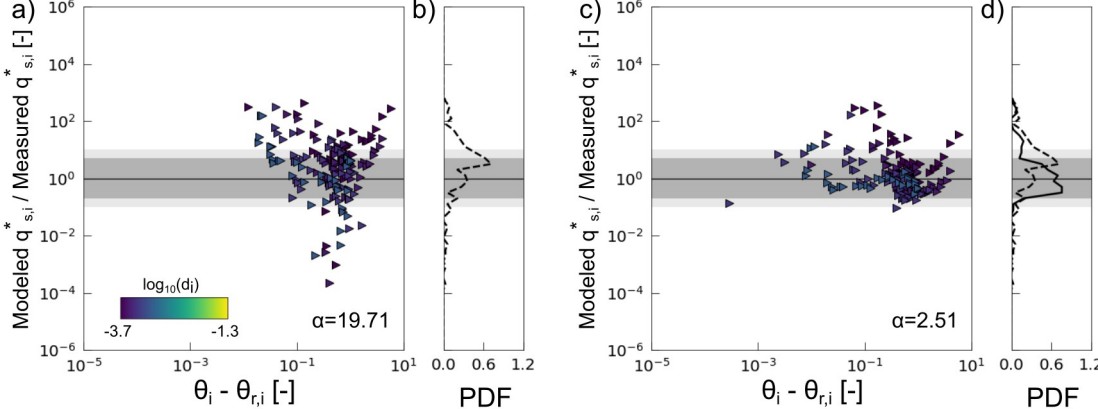

**Figure 6: Effects of standardized incipient motion and adjusted transport height on predictions of total load transport rates at equilibrium for Engelund and Hansen (1967). a-b) Predictions with the same critical shear stress values as of Le Minor et al. (2022) and the original parameterization of LM2022's model ($\alpha = 19.71$). c-d) Predictions with the LM2022's model adjusted with the bed roughness length as reference height across all grain sizes, a bed roughness of $3\,d_{90}/30$ (no bed forms) and the standardized reference shear stress values after re-calibration ($\alpha = 2.51$). The ratio of model predictions to flume observations is plotted against the dimensionless excess of shear stress (Shields stress) along with the probability density function (PDF) of the residuals. For the PDF, the dashed line corresponds to predictions with the original LM2022's model. The solid line in d) corresponds to predictions with the adjusted model. The dark and light gray areas correspond to measured values that are predicted within a factor of 5 and 10, respectively.**

## 5.4 Potential parameters that could improve the model

Despite significant improvements in model predictions, the multiple-size transport rates measured by Shvidchenko et al. (2001) are slightly overestimated compared to the ones of Wilcock and Crowe (2003), suggesting that second-order physical processes are not considered. Here, we explore some of these by examining the effect of the Froude number, the relative depth (ratio between the grain size or the water depth) or the relative roughness (ratio between a characteristic length scale of the bed surface and the grain size), the particle Reynolds number and the fraction of mobile sediment.

De Leeuw et al. (2020) found that the strong correlation of the entrainment coefficient with the Froude number is due to the definition of the Froude number that includes the depth-averaged flow velocity and the water depth, two key determinants of sediment concentration in the bed load layer. Thus, the Froude number that represents turbulent flow properties (Cheng et al., 2020) indirectly characterizes sediment transport and grain mobility. For a low relative depth and thus a high friction coefficient, near-bed turbulence is reduced, and consequently, sediment transport rates are lowered (Lamb et al., 2017b). Near-bed turbulence may also drop due to sediment-induced density stratification, and thus, the entrainment rate is reduced



(Wright and Parker, 2004). For Froude numbers of one or above, surface waves cause turbulence that reaches the sediment
bed and increases sediment mobility and, thus, entrainment. Furthermore, the relative roughness characterizes the grain
protrusion through the bed surface. The bed roughness affects the flow features and the near-bed turbulence intensity due to
grains that protrude through the bed (Vollmer and Kleinhans, 2007).

Table S2 and S3 in the Supplementary Materials explore the effect of including these factors individually or combined in the
entrainment factor, and the corresponding model prediction quality for single and multiple-size datasets. However, we found
no clear evidence that the entrainment factor varies with dimensionless parameters such as the Froude number, the relative
roughness or the Reynolds number. The fact we do not find a strong correlation between these parameters and the
entrainment factor may stem from the concomitance of their effects or simply the considerable dispersion in the data that
may mask weak trends.


For multiple-size sediments, Yager et al. (2007) suggested considering the limited availability of mobile sediment. To
determine the effect of sediment availability on bed load transport in steep boulder bed channels where gravel grains are
mobile, contrary to boulders that rarely are, they carried out flume experiments with sediment beds consisting of mobile
natural gravel grains and spheres mimicking immobile boulders. They showed that the less mobile the bed surface is, the
lower the quantity of sediment available for transport, hindering the mobility of the mobile grains. None of the entrainment
relations published so far, to our knowledge, show a dependency of entrainment on the immobile fraction at the bed surface
in the case of sediment mixtures. While not explicitly considered in existing entrainment relations, existing research attempts
to unravel the potential importance of the immobile fraction by looking at shear stress partitioning (e.g., Gilbert and Wilcox,
2024; Nativ et al., 2022).


We tested the effect of an adjusted entrainment factor that varies with the mobile fraction on the bed surface rather than
being constant. The mobile fraction was calculated as the sum of surface fractions of mobile grain sizes, i.e., reference shear
stress exceeded. Surface fractions of grain size, regardless of their reference shear stress, were included in the mobile
fraction if they had a non-zero measured transport rate. We forced the entrainment factor to be equal to the one calibrated
using the single-size data and the multiple-size data when the bed surface was fully mobile, which resulted in an empirically
fitted entrainment factor $\alpha = 2.51 F_{mobile}^{2.24} = 2.51 \left( 1 - F_{immobile} \right)^{2.24}$. However, the low correlation coefficient (
$R^2 = 0.07$) that may be due to the data's dispersion does not allow us to draw any conclusion about the role played by the
mobile surface fraction in the entrainment rate.

Another important aspect to consider is the dispersion of the sediment transport data, especially for mixed-size sediment,
which makes finding strong correlations challenging. More data are needed to better calibrate the model and to identify the



contribution of experimental conditions on the entrainment factor (especially for grain size sorting above 4). There also could be additional physical complexity such as grain protrusion through the bed surface (Dey and Ali, 2019; Dwivedi et al., 2011; Lamb et al., 2017a; Lee and Balachandar, 2017; Papanicolaou et al., 2002; Xie et al., 2023), grain packing density in the bed (Cheng and Chiew, 1998; Lamb et al., 2017a)and grain shape (Schmeeckle et al., 2007) but our current objective is to have a parsimonious model capturing first order phenomena, which we believe is the case.

## 6 Conclusions

We explored several improvements of the model by Le Minor et al. (2022). First, we introduce a new formalism for the reference shear stress of multiple-size sediments that successfully combines the approaches of Shvidchenko et al. (2001) and Wilcock and Crowe (2003). With a model formulation of intermediate complexity, the reference shear stress model applies to a large range of sediment mixtures and contributes to a significant improvement of LM2022's model when applied to both single and multiple-size datasets. Another important improvement is the definition of a reference transport length set by the bed roughness length. When accounting for these two factors, and by adjusting the mobility factor of the model ($\alpha = 2.51 \pm 28.92$), LM2022 provides greatly improved model predictions for both single and multiple-grain size transport prediction in the bedload regime, and for single grain size suspended load. Further testing of the model is limited by the availability of flume or field data suitable to test it beyond the range of grain sizes, slope and transport stages explored here.

To ensure the quality of predictions with this model, we recommended the use of:
 • a reference shear stress determined with the incipient motion criterion and a reference value of $10^{-4}$;
 • the transport height adjusted with the bed roughness as the reference height across all sizes (Eq. (19));
 • the simplified version of the transport velocity since it does not degrade the transport rate predictions (Eq. (13)).
Seeking a transport model as universal as possible while remaining relatively parsimonious is a matter of compromise. While additional model complexity could certainly be brought in the model to account for effects of partially mobile bed or bed form developments, we consider that at this stage of development, the model is already usable to explore sediment transport and the resulting morphodynamics of rivers under a wide range of hydraulic forcing, median grain sizes and grain size heterogeneity. In particular, using it in fully coupled morphodynamics model to compare with flume experiments studying grain size sorting will provide another range of tests to evaluate its performance.

## 7 Data and code availability

Data tables and Python scripts that support the findings of this study are available in a Zenodo repository (Le Minor, 2025) at: https://doi.org/10.5281/zenodo.15043113.




## 8 Author contribution

MLM: conceptualization, formal analysis, funding acquisition, methodology, validation, visualization and writing – original draft preparation, review & editing.

DL: conceptualization, funding acquisition, methodology and writing – review & editing.

JH: conceptualization, funding acquisition and writing, methodology – review & editing.

PD: funding acquisition and writing – review & editing.

## 9 Competing interests

The authors declare that they have no conflict of interest.

## 10 Financial support

The authors acknowledge: i) the Brittany Regional Council (France) for its financial support of the SAD/SEDRISK project, ii) the New Zealand Governments Ministry of Business, Innovation and Employment's Endeavour fund for its financial support of the Earthquake Induced Landscape Dynamics programme (C05X1709), iii) the ANR – FRANCE (French National Research Agency) for its financial support of the WIVA project n°ANR-21-CE49-0004-02, and, iv) grant from the European Union's Horizon Europe through Marie Sklodowska-Curie Actions (grant no. 101107488).

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
