# Peer review of "Improving a multi-grain size total sediment load model through a new standardized reference shear stress for incipient motion and an adjusted saltation height description"

_EGUsphere, 2025_

## Referee Comment (RC2)

Le Minor and colleagues provide a thoughtful reanalysis of two transport data sets to test and modify their earlier transport model (Le Minor et al. 2022). The two datasets (Shvidchenko et al., 2000, 2001 and Wilcock, Kenworthy, Crowe 2001) have not been analyzed together, as far as I know. Both datasets can be interpreted providing the grain size of the bed surface for the purpose of scaling fractional transport rates – an essential feature. The statistical analysis is clear and credible, as is the evaluation of the model fit. I support publication and my comments are primarily intended to provide perspective and points of interest for the authors in preparing the final version. In particular, I suggest that the authors include further discussion of the *interpretation* and *application* of the model relative to earlier models (basically my own: Wilcock and Crowe 2003 [WC2003]) This is not to say that the present model is inferior, but the choices made in developing the model have implications for its interpretation and application. I believe a discussion of these conceptual differences would be useful for the reader. They should also address the fact that their model in effect defines two different values for the reference stress for $D_{50}$ (Point C below).

**Here are three main points**

(A) **The data used.** The two datasets have some important differences. Shvidchenko used very small transport rates that hardly modified the bed surface. Hence, the bed surface grain size from the start of the run was used to scale the transport rates (although he also demonstrated that the bed surface changed little over each run). Wilcock et al. used a wide range of transport rates (including very low transport rates) and much longer run times. The bed surface grain-size distribution was measured at the end of each run, such that the transport rates of each size fraction could be scaled by its proportion on the bed surface.

In both sets of experiments, the final bed surface and measured transport rate are sensitive to the initial bed grain-size distribution. No sediment was fed in the Shvidchenko experiments and sediment was recirculated in the Wilcock experiments. Parker and Wilcock (1993) show that the flume bed surface and transport are dependent on initial conditions in these cases. Hence, the manner of preparing the bed surface prior to each run becomes important. In both cases, the bed was screeded flat with a blade. This condition is particularly important in the case of the Shvidchenko data because the runs were short and involved very little transport, such that the bed remained relatively unchanged. The initial bed preparation has less of a direct effect on the Wilcock experiments because the runs were much longer (especially at small transport rates). Nonetheless, some of the coarser grains on the bed surface remained immobile throughout the run (the condition of partial transport). I emphasize this sensitivity to the initial, screeded bed because the larger grains on the bed surface were emplaced by the passage of a blade – a different mechanism than depositing from transport. I have no idea what the effect would be on entrainment of coarser grains from a screeded bed vs a water-laid bed, but the effect would be considerably greater in the Shvidchenko data than the Wilcock data. Even though both transport data sets are scaled by the grain size of the bed surface, the entrainment results could be different. As demonstrated in this paper, fitted values of $\tau_{ri}$ for the Shvidchenko data are

consistently larger than those of the Wilcock data. The reason for this difference may well be methodological.

(B) **Conceptual basis of the model.** The author's new model (LM2025) revises their 2022 model. Some discussion of the conceptual differences with the Wilcock/Crowe (WC2003) model could be useful to the reader.

(1) WC2003 uses a single transport function in the determination of the reference stress for each size fraction $\tau_{ri}$. This is for consistency in the model application of $\tau_{ri}$ with the same transport function. LM2025 find $\tau_{ri}$ using a function fitted to each fraction. This has the effect of conceptually separating the reference stress from the transport function. Not wrong, just different.

(2) LM2025 use a contant value of $q*_{bi}$ to define the reference shear stress; WC2003 use a constant value of $W*_i$. Again, not wrong, just different. The effect on the measured values if $\tau_{ri}$ is clear: using a constant $q*_{bi}$ ($10^{-4}$) to define the reference transport rate leads to smaller values of $\tau_{ri}$ for the smallest sizes and larger values of $\tau_{ri}$ for the largest sizes, compared to using a constant value of $W*_I$ (0.002) to define the reference transport rate.

(3) WC2003 builds on previous efforts, dating to Egiazaroff (1965) and Ashida and Michue (1971) which use a similarity collapse to identify a single transport function that applies to all fractions. There is something both profound and convenient to the idea that a *single* transport function applies for all fractions in any sediment mixture, such that all differences in transport between fractions and/or between sediments can be accommodated in terms of the critical or reference shear stress. Although not perfect, decades of work have shown that dimensionless transport rate varies consistently with the excess of shear stress over critical. The LM model is more complex than this and loses this simple interpretation. It is not clear to me that LM2025 is parsimonious compared to models based on a similarity collapse using a universal transport function.

(4) LM2025 test the effect of sorting ($s_g$) and the fraction of sand ($F_s$) on the overall mobility of each sediment mixture (based on the reference stress for the median surface size). They find that $s_g$ provides greater statistical explanation and do not include $F_s$ in the model. What is lost here is the conceptual directness of evaluating the effect of sand on gravel transport, something that has been amply demonstrated over the years (Jackson and Beschta, 1984; Ikeda and Iseya, 1988, Curran and Wilcock, 2005; Hill et al., 2017). WC2003 predicts the transport of gravel, with or without sand. Inasmuch as the composition of a gravel bed may be expected to change much more slowly than the fraction of sand on the bed (which might come and go), it is useful to have a direct means of modeling the effect of sand content on gravel transport. One can, of course, calculate a new $\sigma_g$ with the addition of sand to a gravel bed, but that seems an indirect way of accounting for an important controlling variable. I don't contest the author's statistical analysis but instead suggest that a discussion of the pros and cons of the different models would be relevant to the reader. Also, it is worth noting that the fit between the reference

Shields Number and $\sigma_g$ (Figure 2a) is driven by the fact that all values of $\sigma_g$ for Shvidchenko are smaller than for Wilcock and (as mentioned above) Shvidchenko $\tau_{ri}$ are consistently larger than WC2003 (real or a methodological consequence?).

(C) The proposed hiding function (Eq. 23, Eq. 24; Figure 2) has the unfortunate property that $\tau_{ri}$ does not equal $\tau_{r50}$ at $D_i = D_{50}$. See figure of hiding functions included below. Please explain how this can make sense.

**Comments on specific parts of the text**

**Line xx** (copied from text)
>>R> review comments

**Line 42** there has been, to our knowledge, only one attempt at a continuous transport law that extends from bed load to suspended load for a wide range of flow strengths and sediment grain size distributions introduced by Le Minor et al. (2022)

>>R> There are a number of works on total load that give separate consideration to suspended load and bedload. A particularly thoughtful one considering the separate and combined effects of suspended and bedload is by Dade and Friend (1998).
>>R> The mixed-size data used in this paper to test the model are either entirely bed load (Shvidchenko) or include only low-flying sand suspension at he scale of the coarser components of a gravel bed (Wilcock, Kenworthy, Crowe). So the model presented here is not really tested against robust suspended transport of mixed-size sediment.

**Line 80** Another issue, is that the reference shear stress measurement relates to a specific bed surface grain size composition and hydraulic conditions. The main limitations of existing definitions are that they were established based either on the grain size distribution of the initial bed surface, assuming that surface composition did not vary much between the initial and final run state (Shvidchenko et al., 2001) *or the grain size distribution of the final bed surface averaged over several runs with similar initial bulk sediment* (Wilcock and Crowe, 2003). These may not be comparable and could introduce inconsistencies in model calibration or validation using different datasets, especially for bed load transport predictions.

>>R> It is not just a matter of averaging "over several runs with similar initial bulk sediment", we demonstrated that the surface GSD for each of the five initial mixtures varied little with flow or transport rate (Wilcock, Kenworthy, & Crowe, 2001). We attributed this to a similar degree of kinetic sorting in each run (the runs with smaller transport rates extended for longer periods, such that all runs had greater than a minimum amount of transport and sorting).

**Line 277** When the reference shear stress is not exceeded, we assume that the transport rate is so small that it can be neglected, and thus, the transport rate equals zero.

>>R> In coarser gravel natural systems, most of the transport occurs at stresses smaller than the reference stress.

**Line 374**

Figure 2a shows that there is no clear influence of $d_{50}$ on $\theta_{r,50}$ and that Eq. (22) holds for S2001 and WC2003. Using a functional relationship similar to WC2003, we found that the correlation between $\theta_{r,50}$ and the sand fraction was weaker (

$\theta_{r,50} = 0.049 \exp\left(-0.97 F_{sand}\right)$, $R^2 = 0.30$), so that we do not use $F_{sand}$ subsequently.

>>R> The poorer fit as a function of $F_s$ is due to the fact that the Shvidchenko reference stresses are all larger than the Wilcock reference stresses (by your fitting method). Also, some of the Shvidchenko mixtures have gravel sizes close to 2 mm, such that using a 2 mm boundary between coarse and fine sediment is not meaningful.

**Line 405** The former two formalisms of Shvidchenko et al. (2001) and Wilcock and Crowe (2003) **cannot be shown on the same plot** as our new formalism since they are not equivalent, i.e., different variables and methods were used to parameterize and evaluate the reference shear stress.

>>R> Reasonable choices can be made to allow the different hiding functions to be placed on the same plot. After all, a user of these functions will do exactly that to evaluate them in comparison. I provide a plot below of the Shvidchenko, WC2003, and LM2025 hiding functions. I use $e = 1.06$ for the relation of Shvidchenko (2001); the variation of $e$ is very limited for the data from his experiments; the variation of $e$ for the other data sets examined by Shvidchenko is somewhat larger, but these data are not based on bed surface grain size and should therefore not be relied on. The Wilcock and Crowe (2003) model uses the mean size of the bed surface $D_m$ rather than $D_{50}$ but plotting the function using $D_{50}$ allows a reasonable comparison. We see that the difference between the Shvidchenko and the Wilcock/Crowe hiding functions is not that large in their region of overlap (although small differences in reference shear stress can produce large differences in transport rate). The Le Minor hiding function has the unfortunate property that $\tau_{ri}$ does not equal $\tau_{r50}$ at $D_i = D_{50}$.

[Figure]

Hiding Functions

Legend:
- Shvidchenko 2001 e = 1.06
- Wilcock Crowe 2003 Dm = D50
- Le Minor 2025 sg = 2
- Le Minor 2025 sg = 4
- Le Minor 2025 sg = 6

y-axis: $\dfrac{\tau_{ri}}{\tau_{r50}}$

x-axis: $D_i/D_{50}$

Svidchenko uses $e = 1.06$, which closely matches his experimental data. Wilcock and Crowe model uses the mean size of the bed surface size distribution; $D_{50}$ used here. Grain-size range of plot matches the range used in Svidchenko and Wilcock/Kenworthy/Crowe experiments. Grain-size range shown for Le Minor trends equal to $D_i/D_{50} = \sigma_g{}^2$

**Line 592** Despite significant improvements in model predictions, the multiple-size transport rates measured by Shvidchenko et al. (2001) are slightly overestimated compared to the ones of Wilcock and Crowe (2003), suggesting that second-order physical processes are not considered.

>>R> The entrainment and transport rates of the Shvidchenko and Wilcock data could be genuinely different, based on the much stronger dependence of the Shvidchenko data on the initial condition of the screeded bed.

**Line 620** None of the entrainment relations published so far, to our knowledge, show a dependency of entrainment on the immobile fraction at the bed surface in the case of sediment mixtures.

>>R> almost the entire bed surface was immobile in the data of Shvidchenko. Wilcock and McArdell (1993) explore the extent of fractional immobility (termed partial transport) for the sandiest of the mixtures used by WC2003 and provide a basis for estimating partial transport.

Ashida, K., and Michue, M. 1971. ''An investigation of river bed degradation downstream of a dam.''
    *Proc., 14th Int. Association of Hydraulic Research Congress*, Vol. 3, Wallingford, U.K., 247–255
Curran, J.C. and P.R. Wilcock, 2005, The effect of sand supply on transport rates in a gravel-bed channel,
    *J. Hydraulic Engineering*. DOI: 10.1061/(ASCE)0733-9429(2005)131:11(961)
Dade WB and PF Friend, 1998. Grain-Size, Sediment-Transport Regime, and Channel Slope in Alluvial
    Rivers, The Journal of Geology, 106(6), pp. 661-676)

Egiazaroff, I. V. 1965. ''Calculation of nonuniform sediment concentrations.'' J. Hydraul. Div., Am. Soc. Civ. Eng., 91(4), 225–247.

Hill, K. M., J. Gaffney, S. Baumgardner, P. Wilcock, and C. Paola (2017), Experimental study of the effect of grain sizes in a bimodal mixture on bed slope, bed texture, and the transition to washload, Water Resour. Res., 53, 923–941, doi:10.1002/2016WR019172.

Ikeda, H., and Iseya, F. , 1988. "Experimental study of heterogeneous sediment transport." *Environmental Research Center Paper* No. 12, Univ. of Tsukuba, Tsukuba, Japan.

Jackson, W. L., and Beschta, R. L. , 1984. "Influences of increased sand delivery on the morphology of sand and gravel channels." Water Resour. Bull., 20(4), 527–533.

Parker, G., and Wilcock, P.R., 1993. Sediment feed and recirculating flumes: a fundamental difference, *The Journal of Hydraulic Engineering*, 119(11):1192-1204. Discussion and closure published March, 1995

Wilcock, P.R. and Crowe, J.C., 2003 A surface-based transport model for sand and gravel, *J. Hydraulic Engineering*. 129(2), 120-128.

Wilcock, P.R., Kenworthy, S.T. and Crowe, J.C., 2001. Experimental study of the transport of mixed sand and gravel, *Water Resources Research*, 37(12), 3349-.3358.

Wilcock, P.R. and McArdell, B.W., 1993. Surface-based fractional transport rates: mobilization thresholds and partial transport of a sand-gravel sediment, Water Resources Research, 29(4):1297-1312

---

## Author Comment (AC1)

Dear reviewers,

My co-authors and I appreciate you for the time you spent reviewing our manuscript and providing valuable comments to help us improve the revised version of our paper. We try our best to address every one of the comments.

To do so, we would need some clarification from the second reviewer, Peter Wilcock. In your review, you state in major comment C) that " the proposed hiding function (Eq. 23, Eq. 24; Figure 2) has the unfortunate property that τri does not equal τr50 at Di = D50. See figure of hiding functions included below. Please explain how this can make sense." And that we should "address the fact that [our] model in effect defines two different values for the reference stress for D50".

At the moment, we do not understand this point as Equations 23 and 24 provide a single value of reference shear stress for $D_{50}$. As $\frac{\tau_{r,i}}{\tau_{r,50}} = \left(\frac{D_i}{D_{50}}\right)^{1-\gamma_i}$, when $D_i = D_{50}$, we do have $\tau_{r,i} = \tau_{r,50}$. Below is the figure we obtain when plotting the hiding functions of Shvidchenko et al. (2001, as S2001), Wilcock and Crowe (2003, as WC2003) and Le Minor et al. (2025, as LM2025, new model applied to 3 values of grain size sorting). This figure shows results for the LM2025's model that differ from the ones provided in your review (otherwise the curves for S2001 and WC2003 are similar to yours).

[Figure]

To help us better understand your comment, could you provide us with some clarification on how you proceeded to make the plot of the different hiding functions provided in your review?

Sincerely,

Marine Le Minor

---

## Author Response (AR1)

**Response to reviewers**

Marine Le Minor1,2, Jamie Howarth1, Dimitri Lague2, Philippe Davy2,

1School of Geography, Environment and Earth Sciences, Victoria University of Wellington, Wellington, New Zealand

2University of Rennes 1, CNRS, Geosciences Rennes, UMR6118, Rennes, France

**Cover letter**

Dear Editor,

We thank you and the reviewers for the time spent reviewing our manuscript and providing valuable comments that led to improvements in the revised version of our paper. We tried our best to address every one of the comments and we hope that the manuscript after careful revisions meets your high standards. The authors welcome further constructive comments if any. In this document, you will find all major comments listed in the reviews along with a description of the modifications made to the manuscript. All modifications in the manuscript have been highlighted in the Article-track changes file, and the corresponding lines in the revised manuscript are indicated in red. Here is a summary of the major modifications we have done.

- 1) To ease reading of the section on existing formalisms for reference Shields stress of mixsized sediments, we added a figure that illustrates their differences.
- 2) A second figure was added in the Results section to compare our new model of reference shear stress to these existing formalisms.
- 3) Model assumptions have also been clarified in the section that describes the model.
- 4) The discussion has been extended to include comments on the data used to establish our new model, an assessment of the impact of model assumptions and parameterization on calibration, and, comparisons of our model to other formalisms.
- 5) New references have been inserted to improve the paper context.

We look forward to hearing your decision one you have had time to assess the modifications we have made.

Sincerely,

Marine Le Minor

**Major comments**

**Figures added to section 2**

M. Kleinhans: While technically correct, chapter 2 is in need of some figures showing example shapes of functions in units that are understandable to a larger readership, lest the paper and the online provided code might become a black box to some users. A few graphs are dearly needed in section 2 for a broader readership (like you do in Fig 1 in section 3).

→ We added a figure in section 2 to illustrate the two formalisms S2001 and WC2003 with dimensions (reference shear stress) and without (reference Shields stress) as a function of grain size similar to the graphs presented in Kleinhans and van Rijn (2002).

**Comment and discussion added on assumptions made**

M. Kleinhans: Certain choices (e.g. logarithmic velocity profile, adjusting the saltation height while neglecting saltation roughness) may have an effect on the coefficients in the final set of equations, which is fine but need to be stated and briefly discussed. See detailed comments.

→ We gave more details on the assumptions made regarding the velocity profile and the roughness height and we extended the discussion on how such assumptions may affect our model and especially the calibrated entrainment coefficient that includes the uncertainty of model parameterization.

**References added**

M. Kleinhans: Certain phenomena need better support by references while other references may perhaps be removed (see detailed points).

→ We added/removed references where it was necessary.

**Comment and discussion added on data used**

P. Wilcock: The data used. The two datasets have some important differences. Shvidchenko used very small transport rates that hardly modified the bed surface. Hence, the bed surface grain size from the start of the run was used to scale the transport rates (although he also demonstrated that the bed surface changed little over each run). Wilcock et al. used a wide range of transport rates (including very low transport rates) and much longer run times. The bed surface grain-size distribution was measured at the end of each run, such that the transport rates of each size fraction could be scaled by its proportion on the bed surface.

→ We added a paragraph in the Discussion to emphasize the effect of methodology and especially bed surface preparation on the interpolated reference Shields stress values (lines 578-584).

**Discussion added on the conceptual differences between models**

P. Wilcock: Conceptual basis of the model. The author's new model (LM2025) revises their 2022 model. Some discussion of the conceptual differences with the Wilcock/Crowe (WC2003) model could be useful to the reader.

- (1) WC2003 uses a single transport function in the determination of the reference stress for each size fraction  $\tau ri$ . This is for consistency in the model application of  $\tau ri$  with the same transport function. LM2025 find  $\tau ri$  using a function fitted to each fraction. This has the effect of conceptually separating the reference stress from the transport function. Not wrong, just different.
  - → We added a paragraph in the discussion to highlight this point (lines 588-591) although this is partially explained in section 2.
- (2) LM2025 use a constant value of  $q^*bi$  to define the reference shear stress; WC2003 use a constant value of W\*i. Again, not wrong, just different. The effect on the measured values of  $\tau$ ri is clear: using a constant  $q^*bi$  (10-4) to define the reference transport rate leads to smaller values of  $\tau$ ri for the smallest sizes and larger values of  $\tau$ ri for the largest sizes, compared to using a constant value of W\*I (0.002) to define the reference transport rate.
  - → We added a paragraph in the discussion to highlight this point (lines 592-594) although this is partially explained in section 2.
- (3) WC2003 builds on previous efforts, dating to Egiazaroff (1965) and Ashida and Michue (1971) which use a similarity collapse to identify a single transport function that applies to all fractions. There is something both profound and convenient to the idea that a single transport function applies for all fractions in any sediment mixture, such that all differences in transport between fractions and/or between sediments can be accommodated in terms of the critical or reference shear stress. Although not perfect, decades of work have shown that dimensionless transport rate varies consistently with the excess of shear stress over critical. The LM model is more complex than this and loses this simple interpretation. It is not clear to me that LM2025 is parsimonious compared to models based on a similarity collapse using a universal transport function.
  - → We agree that LM2022's model and its adjusted version LM2025 presented in the revised manuscript do not provide a link between excess of shear stress and transport rate as directly as the similarity collapse used to establish WC2003. However, LM2022's and LM2025's model include a more detailed description of processes driving sediment transport as they are physics-based.
  - → Also, LM2022 is (i) a non steady-state model, and (ii) describing the full range of sediment transport from bedload to suspended-load. It can resolves transient behaviors or bedload and suspended load that WC2003 alone, a steady-state multi-grain bedload model, can

- not. So in the end, while being not as parsimonious as WC2003, it has a broader scope of application.
- → We added a paragraph in the discussion to highlight this point (lines 595-601).
- (4) LM2025 test the effect of sorting (sg) and the fraction of sand (Fs) on the overall mobility of each sediment mixture (based on the reference stress for the median surface size). They find that sg provides greater statistical explanation and do not include Fs in the model. What is lost here is the conceptual directness of evaluating the effect of sand on gravel transport, something that has been amply demonstrated over the years (Jackson and Beschta, 1984; Ikeda and Iseya, 1988, Curran and Wilcock, 2005; Hill et al., 2017). WC2003 predicts the transport of gravel, with or without sand. Inasmuch as the composition of a gravel bed may be expected to change much more slowly than the fraction of sand on the bed (which might come and go), it is useful to have a direct means of modeling the effect of sand content on gravel transport. One can, of course, calculate a new og with the addition of sand to a gravel bed, but that seems an indirect way of accounting for an important controlling variable. I don't contest the author's statistical analysis but instead suggest that a discussion of the pros and cons of the different models would be relevant to the reader. Also, it is worth noting that the fit between the reference Shields Number and og (Figure 2a) is driven by the fact that all values of og for Shvidchenko are smaller than for Wilcock and (as mentioned above) Shvidchenko tri are consistently larger than WC2003 (real or a methodological consequence?).
  - → We agree that LM2025 does not directly account for the effect of sand on gravel transport. However, it includes the effects of sorting which is another way to look at the ratio of fine to coarse sizes and thus pore filling. Besides the statistical significance of sorting compared to sand fraction, LM2025 does not aim only at sand and gravel mixtures but at wider grain size distributions as well and thus sorting instead of sand fraction seems more appropriate to broader applications. We added a paragraph in the discussion to highlight this point (lines 602-606) although this is partially explained in section 2.

**Hiding function checked**

P. Wilcock: The proposed hiding function (Eq. 23, Eq. 24; Figure 2) has the unfortunate property that  $\tau ri$  does not equal  $\tau r50$  at Di = D50. See figure of hiding functions included below. Please explain how this can make sense. [The authors] should also address the fact that their model in effect defines two different values for the reference stress for D50 (Point C below).

ightharpoonup We do not understand this point as Equations 23 and 24 provide a single value of reference shear stress for D50. As  $\frac{\tau_{r,i}}{\tau_{r,50}} = \left(\frac{D_i}{D_{50}}\right)^{1-\gamma_i}$ , when  $D_i = D_{50}$ , we do have  $\tau_{r,i} = \tau_{r,50}$  as shown in the figure below that illustrates the different hiding functions of Shvidchenko et al. (2001, as S2001), Wilcock and Crowe (2003, as WC2003) and Le Minor et al. (2025, as LM2025, new model applied to 3 values of grain size sorting). This figure shows results for the model of Le Minor et al., 2025) that differ from the ones provided by Peter Wilcock in his review.

→ We asked Peter Wilcock for clarification regarding this comment as we do not understand how it can be made from Equations 23 and 24.

→ We added a figure in the Results section (Figure 4 line 438) to illustrate how our new formalism compares with existing ones (both in terms of hiding function and reference Shields stress).

**Detailed comments by M. Kleinhans**

Lines 10 and 36: Why is such catastrophic release relevant for this paper? I can guess, but it would be good to spend a sentence more on this, explaining the importance of armouring and such in the quasicyclic landscape processes described in Tunnicliffe. Adding another reference would nicely contextualize this, for example the review by de Haas et al. (2015) also points at the links between debris flow fans and fluvial fans, armouring and the river valley dynamics. The challenges of linking mountain slope to valley and fluvial plain go far beyond that of downstream fining and indeed require a very universal transport relation suitable for extremely wide grain size distributions.

→ We added a reference and a sentence at the end of the first paragraph in the introduction to better set the context behind the development and adjustment of a multi-grain size total sediment load model (lines 37-39).

Line 43: I do not agree prima vista, because it all depends on what one calls continuous. One can argue that van Rijn 1984 is sufficiently continuous because it honours the physical difference in sediment motion modes (saltation vs suspension) but uses the bedload relation as a basis for the suspended load relation because the saltation layer is where the suspended sediment originates.

In fact, Le Minor et al. 2022 use the saltation height (here too in eq. 11). So this points at a question: why should it be more continuous in the sense meant by the authors when the physical phenomena are in fact not more continuous? This needs a convincing argument, or a brief explanation why this concept is equally interesting as the one used by van Rijn and others.

→ We agree that a number of works exist on total load but the concept differs from the continuous transport law of Le Minor et al. (2022) in that bedload and suspended load are calculated separately and summing both provides the total load. In other words, Le Minor et al. (2022) do not discriminate between bedload and suspended load as a single formalism is used to represent both as well as the transition from one to the other. We added sentences at the end of the second paragraph of the introduction to state that other approaches exist to determine total load (lines 47-54).

Lines 65-85: The Parker approach for reference transport and his notion that it is the surface layer composition that matters, not the bulk, is missing from this otherwise insightful review. Parker (1982, 1993) argues, and presents some evidence, that the hiding-exposure phenomenon and the armouring (and as shown by Blom as cited and myself in 2005, bedforms) are linked intricately. In this sense the system is complex (with feedbacks) rather than complicated.

→ We added a sentence to describe that point in the introduction (lines 94-96).

Line 98: At this point the term transport length has been dropped several times, and I know it is the topic of Davy & Lague 2009, but the reader needs an explanation earlier how this transport length relates to sediment transport as the volume per unit area per second that most are familiar with. I propose to state this early, and its relation to Shields number, because then it is obvious why the entire section 2 is about that number. Now the relation only comes in 275 eq. 18, which does not help readability. If only a functional relation is provided that would already help, for example the inverse of equation 21 (solved for q\_s) or something. Or provide parts of section 2.2 first, for example line 241-244 explains part of the story already.

→ We added two sentences earlier in the text to explain the term transport length (lines 60-64).

Table 1: The reference to the empirical equation of Nielsen is unclear. This is a book on coastal boundary layers and sediment transport and Nielsen worked on coastal bedforms. Not only is it unclear which equation is used here, but also is it unclear why this had to replace Soulsby's equation. This deserves a place at least in the supplementary information.

→ We tried three equations of bed roughness height to test the sensitivity of the model to bed roughness as it is used in the logarithmic velocity profile in Le Minor et al. (2022). This was mostly about the dataset collected by Guy et al. (1966) and used by Engelund and Hansen (1967) to establish their total load transport law. For LM2022, we found that predictions (model vs observations) were better within a factor 5 and 10 when using the equation of Nielsen (1992). For dataset with no information on bed roughness nor bedform

dimensions, we decided to use the Soulsby equation that is a simple relationship between bed roughness and characteristic grain size. So we added a sentence in the caption of Table 1 to refer the reader to Le Minor et al. (2022) regarding the bed roughness equation used (lines 137-140).

Line 173: Perhaps mention that d50 is in mm in this equation, hence the factor 1000.

→ In Equation 8 in Shvidchenko et al. (2001), there is a factor 1000 in front of the d50 expressed in meters. As we have not changed the units of d50 from those used by the original authors, we did not modify the text (line 192).

Line 165-196: Yes, complicated (not complex) indeed, and the reader certainly needs a figure here showing the resulting theta\_r,i and the shields curve not accounting for hiding effects for the different methods (for example as my fig 1-3 in my 2002 paper). Please make such a figure for a sediment in your dataset, or multiple figures if they cannot be plotted together.

- → We replaced the term "complex" by "complicated" in the text and added a new figure that shows such comparison of reference shear and Shields stress values with and without hiding effects (line 224).
- → We illustrate in the new Figure 1 (line 204) the effects of hiding-exposure on the reference Shields stress using the grain size distribution of the sediment mixture with the smallest sand fraction in WC2003's dataset as it falls within the sorting range for which S2001's was also established. Two sentences were added to the text to refer to this new figure (lines 201-202 and 221-222).

Line 245: Mention that this assumes the law of the wall, which is fine, but differs from the linear velocity profile derived from measurements in gravel beds and the double-averaged Reynolds equations, which we also assume between the roughness elements (Vollmer Kleinhans as cited) and therefore would apply to all sediment sizes below D84. Assuming a log profile means that deviations are ascribed later to the hiding or grain-size dependent entrainment model. Please explicate this here or elsewhere. Furthermore, this equation assumes that the roughness is z\_0, while we have indications that the saltation layer thickness actually adds to the roughness (Kleinhans et al. 2017). Assuming this is not the case also affects the empirical coefficients later.

→ We clarified the assumption of the law of the wall when calculating the transport velocity in the text (lines 270-271). We agree that assumptions made in model parameterization have effects that propagate and are ultimately accounted for in the entrainment coefficient that is the only parameter that needs calibration against observations. We added a paragraph to discuss how model assumptions may affect the entrainment coefficient (lines 643-650).

Line 285: On a similar note, this adjustment could also be necessitated by a deviation from the logarithmic velocity profile. Furthermore, it could be necessitated from the three-dimensional structure of the grains on the bed surface as shown by Kirchner et al. 1990 and shown to have a huge effect on the hiding function by Kleinhans & Vollmer (2008 we were on a similar path as you here but family matters took over and we never published this as a paper so I am happy you are doing something like this now).

All this is fine, we know the system is complex and we need to neglect certain feedbacks otherwise we have an underdetermined mathematical system to fit on limited data, so I think I understand sufficiently that, and why, you make these choices, but they need to be stated and made clear to the reader. Also, this needs to be discussed.

→ We added discussion on the error propagation to explain that assumptions we make have consequences on other parameters, especially on the only calibrated one that is the entrainment coefficient (lines 643-650).

Fig 2a: A power function was fitted here (which makes sense) but was the condition of homoscedasticity valid enough for this data (normal distribution of points on a double log scale)? If not, are those high sigma cases affecting the function very much?

ightharpoonup We found that the sorting values are not normally distributed contrary to the reference Shields value (tested in Python). The low number of interpolated reference Shields stress values for the median grain size does not really allow to draw statistical significance regarding the distribution. However, we tested the effect of high sorting values that correspond to WC2003's data. With WC2003's data,  $\theta_{r,50}=0.060~\sigma_g^{-0.469}$  and without  $\theta_{r,50}=0.051~\sigma_g^{-0.742}$ . So high sigma values associated with WC2003's data affect significantly the scaling exponent of the reference Shields stress for the median grain size with the sorting. We added a sentence in the text to state that more data with high sorting would help to better constrain our new model of reference Shields stress for wide grain size distributions (lines 402-404).

Line 376: I am relieved that F does not matter that much, or I would be inclined to ask why the entirely artificial 2 mm distinction between sand and gravel was used rather than a geometrical measure such as the pore-filling size (Frings et al. 2008). (Don't take this point too serious.)

→ As we aim at developing a model suitable for a wide grain size distribution, we argue that sorting becomes a better indicator of poor filling than sand fraction once you have very wide distributions (not only sand and gravel), although sorting is a less direct way to quantify pore filling and accounting for near-bed hydraulic effects. We added a sentence on the text to discuss sorting and sand fraction as proxies for pore-filling size (lines 541-542).

Line 388: I would prefer to have Fig S2 in the paper, possibly expanded with earlier equations referred to here so that it is visualized for the reader.

→ We moved Figure S2 from the Supplementary File into the paper and added equations of fitted curve for better visualization (Figure 3, line 429). We did not add curves corresponding to hiding functions for S2001 and WC2003 to plot b) because, for S2001, the exponent only depends on sorting (and d50 as well) for di/d50 smaller than one while, for WC2003, the exponent does not depend on sorting. For plot c), as we show the dependence of the hiding exponent to di/d50 remaining after dependence on sorting was removed, it is not possible to compare to existing hiding functions.

Lines 391 and 523: Rather than reporting R^2, it might be better to report significance because with a large number of points a low R^2 may still be significant. As you also state "Our model is slightly more complex than Wilcock and Crowe (2003) but has lower residuals", it makes sense to calculate the significance which would account for the larger number of variables in your model.

→ To complement the R-squared values obtained from Pearson's correlation we have added the associated significance as p-values in the text (lines 423 and 448-450). Despite low R-squared for S2001 and WC2003, there is indeed a significant correlation between predicted values of reference Shields stress and interpolated ones.

Fig 3: It is far too small and a colour graph would be better. This is an important graph and a nice result.

→ We modified the figure by making it bigger and in color (Figure 5 line 452).

Line 446: And the strong trend (fig. 4a) is reduced (compare fig 5a).

→ This attenuation of the trend is already commented in the text (lines 472-474) and as the recalibration of the entrainment factor does not affect the trend, we did not modified the text.

Line 498: To be fair to Shvidchenko, while he did not fit a function he certainly considered the possible dependence on the grain size distribution, because he designed his tests with narrow and wide distributions and with skewed distributions. I am, in fact, quite certain that a future expansion to your functions should include skewness, because a skewed distribution puts the various grain sizes at different elevations (Kirchner's concept but not quantified by him, see Kleinhans and Vollmer 2008) which must affect their hiding function.

→ We modified the text to clarify that Shvidchenko et al. (2001) considered a dependence on sorting through the various grain size distributions of experiments they conducted (lines 537-538). We added a sentence on the use of skewness as another parameter that could be considered to improve the model in the discussion (lines 576-577).

Line 511: But also the pressure fluctuations into the bed which may entrain the finest sediments (Vollmer) and angularity differences between size classes and so on, and a hiding function hides/parameterizes effects of changes in the boundary layer structure (Vollmer) and the turbulence forces on the different grain weights (Kleinhans & Van Rijn). Better state here than in final paragraph of the discussion.

→ We moved the text on that point earlier in the discussion and added the effects described in the comment (lines 555-558).

Line 549: This needs a reference, and I suggest the elegant work of Ferguson (2007, 2012).

→ We added the provided reference in the text (line 624).

Line 577: Not only roughness, but also density effects on turbulence damping as you also state in line 604. Better state here I think. EH's data is of fine sediment at high Shields number so I wonder whether this was important (Wright and Parker use a modified Engelund Hansen relation for density effects if I remember correctly). See, for example, Van Rijn 2007 and Van Maren (2009 and earlier refs therein) for larger effects of density. This might well be relevant for catastrophic overloading of rivers and mudflows.

→ We added a sentence in the text that details the relevance of density effects in the case of catastrophic sediment release to rivers (lines 663-665).

Line 598: Really De Leeuw? this finding goes back to the 1960s, for example Vanoni also expressed various things as a function of Froude number.

→ We changed the reference to Vanoni (1974, line 685).

Line 604: A reference to Winterwerp (2001) is deserved who worked on this long before Wright and Parker.

→ We agree and thus added the reference to the text (line 664).

Line 607: Not only near-bed turbulence, but also in-bed pressure fluctuations.

→ We added this effect as it was lacking (line 556).

Line 616: Not only Yager et al but also earlier work by Carling, Ikeda Seal-Paola, and others including myself (Kleinhans et al 2002 and refs therein).

→ We modified the sentence and added relevant references (lines 707-709).

Line 622: Kleinhans and Van Rijn 2002 explicitly considered hindering in their semi-empirical sediment transport predictor. I acknowledge it is primitive.

→ We added this reference to the text (lines 708-709).

Line 680: The online repository shows a seemingly different European funding scheme or project, which also explains why you mention the landslides early in this paper.

→ We added a sentence in the introduction to better explain the context of the paper that is included in this project (lines 37-39).

Table S3: Multiple variables in the last columns deserve a better visualization. Perhaps multicolumn on a landscape page?

→ We modified the table for better visualization of results by extending the number of columns on a landscape page.

**Detailed comments by P. Wilcock**

Line 42-44: There are a number of works on total load that give separate consideration to suspended load and bedload. A particularly thoughtful one considering the separate and combined effects of suspended and bedload is by Dade and Friend (1998).

→ We agree that a number of works exist on total load but the concept differs from the continuous transport law of Le Minor et al. (2022) in that bedload and suspended load are calculated separately and summing both provides the total load. In other words, Le Minor et al. (2022) do not discriminate between bedload and suspended load as a single formalism is used to represent both as well as the transition from one to the other. We added sentences at the end of the second paragraph of the introduction to state that other approaches exist to determine total load (lines 47-54).

The mixed-size data used in this paper to test the model are either entirely bed load (Shvidchenko) or include only low-flying sand suspension at the scale of the coarser components of a gravel bed (Wilcock, Kenworthy & Crowe, 2001). So the model presented here is not really tested against robust suspended transport of mixed-size sediment.

→ We agree that the mode of transport reported for the multiple-size sediment data used to test the adjusted model is mostly bedload. However, to cope with this limitation, we also tested our model against single-size sediment data collected by Guy et al. (1966) and used by Engelund and Hansen (1967) on sand transport that includes bedload and suspended

load as illustrated in Figure 8. We added two sentences to clarify that point in the introduction (lines 725-729).

Lines 80-85: It is not just a matter of averaging "over several runs with similar initial bulk sediment", we demonstrated that the surface GSD for each of the five initial mixtures varied little with flow or transport rate (Wilcock, Kenworthy, & Crowe, 2001). We attributed this to a similar degree of kinetic sorting in each run (the runs with smaller transport rates extended for longer periods, such that all runs had greater than a minimum amount of transport and sorting).

→ We added a sentence to clarify that point in the introduction (lines 96-102).

Line 277: In coarser gravel natural systems, most of the transport occurs at stresses smaller than the reference stress.

→ LM2022 and LM2025 rely on the concept of a threshold of motion and thus it is a choice to have no transport for bed shear stress below the reference shear stress. We justify this choice by the very low transport observed when the reference shear stress is not exceeded as less than one grain for ten thousand available at the bed surface are transported per second (transport intensity criterion corresponding to weak transport as explained by Kramer, 1965). We modified a sentence in the text to emphasize this conceptual difference between our model and existing transport laws (lines 306-307).

Line 374: The poorer fit as a function of Fs is due to the fact that the Shvidchenko reference stresses are all larger than the Wilcock reference stresses (by your fitting method). Also, some of the Shvidchenko mixtures have gravel sizes close to 2 mm, such that using a 2 mm boundary between coarse and fine sediment is not meaningful.

→ We agree that using the sand fraction as a proxy for pore-filling size for S2001's dataset is not the best. However, as we aim at developing a model suitable for a wide grain size distribution, we argue that sorting becomes a better indicator of poor filling than sand fraction once you have very wide distributions (not only sand and gravel), although sorting is a less direct way to quantify pore filling and accounting for near-bed hydraulic effects. We added a sentence on the text to discuss sorting and sand fraction as proxies for pore-filling size (lines 541-542).

Line 405: Reasonable choices can be made to allow the different hiding functions to be placed on the same plot. After all, a user of these functions will do exactly that to evaluate them in comparison. I provide a plot below of the Shvidchenko, WC2003, and LM2025 hiding functions. I use e = 1.06 for the relation of Shvidchenko (2001); the variation of e is very limited for the data from his experiments; the variation of e for the other data sets examined by Shvidchenko is somewhat larger, but these data are not based on bed surface grain size and should therefore not be relied on. The Wilcock and Crowe (2003) model uses the mean size of the bed surface Dm

rather than D50 but plotting the function using D50 allows a reasonable comparison. We see that the difference between the Shvidchenko and the Wilcock/Crowe hiding functions is not that large in their region of overlap (although small differences in reference shear stress can produce large differences in transport rate). The Le Minor hiding function has the unfortunate property that  $\tau ridoes$  not equal  $\tau r50$  at Di = D50.

Svidchenko uses e=1.06, which closely matches his experimental data. Wilcock and Crowe model uses the mean size of the bed surface size distribution;  $D_{50}$  used here. Grain-size range of plot matches the range used in Svidchenko and Wilcock/Kenworthy/Crowe experiments. Grain-size range shown for Le Minor trends equal to  $D_f/D_{50} = \sigma_g^2/2$

→ See response to major comment "Hiding function checked". We added a figure in the Results section to illustrate how our new formalism compares with existing ones (both in terms of hiding function and reference Shields stress) and commented on it in the text (Figure 4 line 438 and text lines 444-445).

Line 592: The entrainment and transport rates of the Shvidchenko and Wilcock data could be genuinely different, based on the much stronger dependence of the Shvidchenko data on the initial condition of the screeded bed.

→ See response to major comment "Comment and discussion added on data used".

Line 620: Almost the entire bed surface was immobile in the data of Shvidchenko. Wilcock and McArdell (1993) explore the extent of fractional immobility (termed partial transport) for the sandiest of the mixtures used by WC2003 and provide a basis for estimating partial transport.

→ We acknowledge that previous studies have looked at partial transport. However, our definition of the mobile fraction indicates that we classify a grain size as mobile if its reference shear stress is exceeded. In this context, the surface for S2001's dataset was fully mobile, in contrast to WC2003's dataset. The mobile fraction for S2001's data was

close to one as we do not consider that only a small fraction of the grains that have their reference shear stress exceeded are actually transported.